# Extended coverage of human serum glycosphingolipidome by 4D-RP-LC TIMS-PASEF unravels association with Parkinson's disease

Huong Giang Vo [1], Gabriel Gonzalez-Escamilla [2,3], Daniela Mirzac[2], Lilia Rotaru[4], Damian Herz[2], Sergiu Groppa [2,3] & Laura Bindila [1] ✉

Glycosphingolipids (GSLs) are important targets in immune, infectious, lysosomal storage diseases, cancer, and neurodegenerative diseases. Circulatory GSLs profiling in clinical samples is restricted by the lack of mid- and high-throughput analytical methods and deep coverage of long-chain sialylated glycosphingolipidome. We present a 4-dimensional (4D)-glyco-sphingolipidomics platform for routine glycosphingolipidome profiling encompassing: extraction and fractionation of sialylated GSLs with 3 to 15 monosaccharides, neutral GSLs and sulfatides; μL-flow reversed-phase LC-TIMS-PASEF MS analysis; semi-quantification strategy adapted for fractionated glycosphingolipidome, and referential CCS, RT, and *m/z* values for GSLs annotation. 4D-glycosphingolipidomics of human serum reveals a high structural heterogeneity, amounting to 376 GSLs: 159 GSLs of ganglio- and neolacto-series, 145 neutral GSLs and 72 sulfatides. Here we demonstrate the platform's utility for clinical profiling of Parkinson's disease (PD) sera. 41 neolacto- and ganglio-species discriminate PD patients from controls and 14 GSLs differentiate sex subgroups, laying the foundation for further functional GSL studies with PD.

Glycosphingolipids (GSLs) consist of carbohydrate chains bound to hydrophobic ceramides and are located on the surface of the cellular membrane[1]. They are essential for cell membrane formation, cell–cell recognition, signaling, immune responses, neuronal functions, and host-pathogen interactions[2]. The disruption of vital cellular processes caused by complex chemical reactions at the molecular level is a key factor in the vast majority of human diseases. Understanding the structure-function relationship of biomolecules can direct the development of effective treatments to prevent or cure diseases. The fact that specific GSL structures change in diseases such as lysosomal disorders[3], infection[4,5],

neurodegeneration[6,7], cancers[8–12], etc. emphasizes the importance of structural characterization and level determination of individual GSLs in biospecimens to comprehend their mechanistic role. However, our current understanding of biological and disease mechanisms involving GSLs remains incomplete because of the challenges presented to routinely extract, profile, and quantify them in clinical and biological samples. GSLs exhibit extensive structural diversity and isomerism arising from variations in: (i) monosaccharide composition[13,14] (Fig. 1a), carbohydrate chain length, isomerism due to branching and glycosidic linkage type and position, possible monosaccharide modification by acetylation,

[1]Clinical Lipidomics Unit, Institute of Physiological Chemistry, University Medical Center, Mainz, Germany. [2]Movement Disorders, Imaging and Neuro-stimulation, Department of Neurology, University Medical Center, Mainz, Germany. [3]Department of Neurology, Saarland University, Saarland University Hospital, Homburg, Germany. [4]Laboratory of Functional Neurology, Diomid Gherman Institute of Neurology and Neurosurgery, Chisinau, Republic of Moldova. ✉e-mail: bindila@uni-mainz.de

sulfation, and other, and (ii) heterogeneous ceramide compositions. Additionally, the neutral, sialylated, and sulfated GSLs, respectively, have different physico-chemical properties that hinder the efficient extraction and analysis of all these GSL categories from a single sample. Their dual hydrophilic-hydrophobic nature makes it difficult to define single sets of extraction and instrumental parameters to encompass all GSLs in a single analysis. Typically, for precise identification of functional activities of individual GSLs in biology and disease, glycosphingolipidomic strategies necessitate the development of GSL subclasses-specific extraction methods and multi-analysis to achieve complete structural heterogeneity profiling, including isomers separation and fine structure elucidation.

**Fig. 1 | 4D-glycosphingolipidomics using RP-LC-TIMS-PASEF MS for glycosphingolipids analysis. a** Symbol Nomenclature for Glycan[13,14] (https://www.ncbi.nlm.nih.gov/glycans/snfg.html) used throughout figures: Glucose (blue circle), Galactose (yellow circle), *N*-acetylglucosamine (blue square), *N*-acetylgalactosamine (yellow square), *N*-acetylneuraminic acid (purple diamond), and Fucose (red triangle); **b** Base peak chromatogram showing separation of glycosphingolipids (GSLs) in customized standard mixture using reversed-phase liquid chromatography. A customized standard mixture was prepared from commercial standards of GM4, GM3, GM2, GM1, GD3, GD2, GD1a, GD1b, GT1b, and GQ1 species in an equimolar concentration of 5 pmol/μL, with 20 μL injected onto the column. GSLs were separated based on their ceramide moiety and degree of sialylation; **c** Partial separation of GD1a 18:1;O2/18:0 and GD1b 18:1;O2/18:0 in extracted ion chromatogram (EIC) and in extracted ion mobilogram (EIM) of *m/z* 917.48 in negative ion mode and of *m/z* 919.50 in positive ion mode. The PASEF spectra obtained from the distinct regions of the partially separated extracted ion chromatogram and mobilogram in negative ion mode demonstrate the differential pattern of the diagnostic fragment ions at nominal *m/z* 581 characteristic for GD1b species, at higher abundance at CCS value of 456.4 Å². In positive ion mode, the disialo element was exclusively found in the spectrum corresponding to the mobilogram at CCS value of 473.5 Å². In both spectra of GD1a and GD1b in positive ion mode diagnostic fragment ions of the 18:1;O2 sphingoid base and the accompanying neutral loss of fatty acyl from the ceramide moiety ions are underscoring the ceramide structure; **d** Three-dimension (*m/z*, collisional cross section (CCS), retention time (RT)) distribution of GSLs in customized standard mixture (average value of each descriptor from *n* = 3 measurements) (Supplementary Data 1); **e** Two-dimension (*m/z*, CCS) distribution of GSLs in customized standard mixture (average value of each descriptor from *n* = 3 measurements). Source data are provided as a Source Data file.

Mass spectrometry (MS) has significantly advanced the GSLs analysis by increasing throughput and sensitivity for structural elucidation compared to other biochemical tools. To this end, MS methods specifically tailored for GSLs, such as glycan-based separation on normal and/or HILIC phase chromatography coupled to high-resolution MS or MALDI MS profiling, have been developed and successfully implemented for GSLs analysis from a variety of biological matrices[12,15–21]. Unfortunately, the sensitivity of the current untargeted lipidomic methods is not conducive to the detection of low-abundant GSLs in a total lipidome extract. In fact, most untargeted lipidomic studies report mainly the hexosyl- or lactosylceramides, and few monosialoganglioside species, such as monosialodiahexosylganglioside (GM3) in a bulk lipidome extract. This is also a primary reason for the lack of experimental MS/MS spectra and ion mobility values for GSLs in lipid databases that are necessary for their identification with high-end ion mobility and MS platforms. Moreover, a variety of elaborated extraction protocols are typically needed to enrich specific GSL subclasses from complex tissue, cells, and blood matrices; more often than not, requiring days or weeks-long and costly wet lab procedures. Removing unwanted matrix effects and enriching GSLs using solid-phase extraction (SPE) has been recently conducted by Horejsi et al. to improve the sensitivity of GSL detection in plasma via MS[17]. Detailed GSLs structural profiling, including Lewis species such as Leᵃ, Leᵇ, Leˣ, Leʸ, and sialyl-Leᵃ/Leˣ from tissues required more than 9 days and several stages of enzymatic treatments and fractionation[22]. Porter et al.[23] summarized the protocols that are useful to guide prospective developments for GSL extractions. Recently, enrichment of neutral GSLs from 100 μL plasma by titanium oxide and charge tagging Paternò-Büchi derivatisation, and reversed-phase liquid chromatography quadrupole time-of-flight MS analysis enabled the structural characterization of 319 neutral GSLs of Hex-, Hex2- Hex3- and HexNAcHex3Cer classes and 32 sulfatides, including the determination of double bond position[24]. The detection and structural profiling of sialylated GSLs in plasma/serum remains more challenging, particularly with respect to longer chain sialylated GSLs with or without modifications by Fuc-, *O*-Ac, Neu5Gc, and at a throughput amenable to clinical samples profiling. To improve the throughput of GSLs profiling in the context of health and disease in clinical samples, faster extraction protocols of the complete glycolipidome amenable for parallelization are required in conjunction with high-end MS capable of deep structural coverage and resolution and reproducible GSLs quantitative profiling.

Ion mobility spectrometry coupled with MS (IMS-MS) emerged as a powerful technique for studying GSLs[25–27]. Structure of lossless ion manipulation, one of the recent IMS developments, separates effectively GSL isomers such as a-series disialotetrahexosylganglioside (GD1a) and b-series disialotetrahexosylganglioside (GD1b)[28]. IMS-MS not only measures the mass-to-charge (*m/z*) ratio of precursor and fragment ions for molecular identification but also separates ions based on their structural conformation. IMS methods using cyclic IMS instruments have been reported to efficiently resolve complex GSL mixtures in three dimensions: *m/z*, ion mobility (*1/K_O*), and MS/MS[29]. Imaging MS using ion mobility gained appeal in profiling the distribution of GSLs, including isomers across tissues[26], showing the power of ion mobility resolution to delineate fine structural isomers at high sensitivity and, ultimately, to pinpoint the biological role such isomers play in functional tissue subregions with diseases. Trapped ion mobility spectrometry (TIMS) with parallel accumulation-serial fragmentation (PASEF) fragmentation[30] has been recently used for lipidomics profiling but not yet for glycosphingolipidomics[31–34].

Hexosylceramides and a few of the GM3 ganglioside species are the main species that could be routinely detected with the 4D-lipidomics and/or with other 3D-lipidomic platforms in bulk lipidome extracts[32,35,36]. Earlier work on nanoflow LC with aqueous normal phase separation/nano-electrospray ionization quadrupole time-of-flight (QTOF) demonstrated great separation of fine structural GSL isomers deriving from: linkage types, *O*-acetylation, *N*-glycolyl-versus *N*-acetyl- substitution of sialic acids, branching, and linkage position of monosaccharides in the glycan chains[12,19,37]. Relative quantification of individual GSLs was challenged with these methods by overlapping isotopologues, requiring extensive deconvolution processing or multiplexed analytical approaches. Besides, GSL standards are only currently increasingly available, but still not for all (sub)classes.

Consequently, the determination of precise biological activities of GSLs in diseases and conditions, where medium to large clinical sample cohorts need to be routinely profiled, is limited for the reasons outlined above.

Based on these prior works, we expect that coupling LC with TIMS-PASEF would enable a superior resolution of glycosphingolipidome based on the four dimensions (4D): *m/z*, retention time (RT), collisional cross section (CCS), and MS/MS spectrum. We hypothesize that reversed-phase (RP) chromatography would enable delineation of isotopologues of heterogeneous ceramide content and a higher separation efficiency and relative quantification of individual GSL species without additional processing steps for monoisotopic peak extraction. We further rationalized that RP-based chromatographic separation of ceramide content would complement IMS-based glycan separation, and along with the operation at μL-flow rates, the RP-LC-TIMS-PASEF would expedite robustness and sensitivity of GSLs profiling and the depth of structural elucidation at higher throughput than before.

In this work, we present an optimized mid-throughput method for the extraction of GSLs from human serum coupled with a sensitive RP-LC-TIMS-MS method for qualitative and semi-quantitative GSLs profiling. This 4D-glycosphingolipidomics platform using RP-LC-TIMS-MS enables in-depth structural elucidation of glycan chain architecture and ceramide composition of distinct GSL species of neolacto- and ganglio-series, enhancing their subsequent semi-quantification in complex samples. By addressing the present limitations of GSLs profiling, this development accelerates the application for clinical

samples. Altogether, 159 sialylated GSLs covering neolacto- and ganglio-series, including long-chain fucosylated, tetrasialylated, and *O*-acetylated species, were detected in human serum, highlighting the remarkable diversity of human serological sialylated glycosphingolipidome and its implications in health and disease. Additionally, 145 neutral GSLs along with 72 sulfatides were detected, amounting to a total of 376 GSLs in human serum.

We present additionally a pilot study of Parkinson's Disease (PD), which demonstrates the robustness and versatility of the 4D-glycosphingolipidomics platform to expedite the uncovering of GSLs fingerprints with diseases in human serum. It also points towards a specific role of the neolacto-series in PD. Besides, serological GSL fingerprints of the sialylated neolacto- and ganglio-series were evidenced to reliably discriminate PD from controls and phenotype PD-groups.

## Results

### 4D- glycosphingolipidomics using RP-LC-TIMS-PASEF

GSLs are typically analyzed using glycan-based separation techniques such as aqueous normal phase or HILIC chromatography[15,17,18,21]. However, using this separation, the processing data for GSLs quantification is tedious because of overlapping isotopologues of coeluted GSLs with similar glycans but different fatty acyls in ceramides: GM3 18:1;O2/18:0 and GM3 18:1;O2/18:1, and/or with different sphingoid bases: GM3 18:0;O2/18:1 and GM3 18:1;O2/18:0. Overlapping extends when multiple ion forms and/or in-source decay species are formed. Using RP-LC, GSLs are expected to separate due to the hydrophobic ceramide moiety. Yet, it was at first not apparent that ionization and detection yields of sialylated GSLs and separation efficiency could enable a broad coverage when using RP-LC. We nevertheless explored the possibility to separate sialylated GSLs based on the ceramide part to avoid isotopologue overlap, increase semi-quantification coverage of individual species, and, concurrently, expand the coverage of the long-chain sialylated glycosphingolipids.

GSL standards (Supplementary Table 1) were used to assess and adapt the previously developed 4D-lipidomics LC-TIMS-MS method[32] for the analysis of GSLs in negative ion mode. Switching off the online calibration substantially improved the ionization efficiency and sensitivity of GSL detection. Switching off the TIMS did not result in any improvement in the sensitivity of detection. On the contrary, to our surprise, the sensitivity of detection increased with TIMS on (Supplementary Fig. 1). The reason for the superior detection sensitivity with TIMS on is not yet clear and can only be at this point potentially attributed to ion accumulation and high ion usage of the TIMS as per instrument provider[38].

The predominant ion types produced for GM3 and monosialotrihexosylganglioside (GM2) are [M−H]⁻. For disialoganglioside (GD), trisialoganglioside (GT), and tetrasialoganglioside (GQ) the ion type is [M−2H]²⁻, for pentasialogangliosie (GP) is [M−H₂O−2H]²⁻, and for monosialotetrahexosylganglioside (GM1) is [M + HCOOH−2H]²⁻. The RP-LC separation of GSLs was based on factors like ceramide moiety length, saturation and hydroxylation, glycosylation and sialylation degree (Fig. 1b). Similar to other lipids, an increase in fatty acyl chain length of ceramide results in greater hydrophobicity and a later elution time, while the opposite is true for shorter chains. A higher number of double bonds in the ceramide base leads to faster elution times. An increase in sialylation and/or glycosylation enhances molecule hydrophilicity and reduces elution time. TIMS was able to partially separate GD1a and GD1b isomers (Fig. 1c), a result consistent with MALDI-TIMS-based recent findings[26]. This isomer separation was confirmed in negative and positive ion mode, whereby protonation of the GD1 increases conformational differences between isomers (Fig. 1c). These isomers are partially chromatographically separated using our global lipidomics RP-LC gradient[32]. Further tailoring of the LC conditions to favor baseline GSL isomer separation due to linkage type and glycan position isomerism is certainly needed.

The RP-LC-TIMS-PASEF fragmentation in negative and positive ion mode was instrumental to ascribe the ceramide composition to specific glycan structures. For sialylated GSLs able to form positive ions, typically GM and GD, the PASEF renders ions generated by ceramide fragmentation to sphingoid base and neutral loss of fatty acyl, as exemplified for GD3 39:1;O2 (Supplementary Fig. 2). In negative ion mode the same species will form exclusively ions by glycosidic bond cleavages (Supplementary Fig. 2). Hence, we can obtain a deeper insight into the diversity and isomeric nature of ceramides (Supplementary Data 1). Surprisingly and noteworthy is that with the RP-LC-TIMS-MS, each subclass of GSLs occupies a distinct region in the 3D space of *m/z*, RT, and CCS (Fig. 1d). In fact, in the *m/z*-CCS dimension, each subclass of GSLs is near baseline delineated (Fig. 1e), which suggests that not only IMS can serve as an orthogonal separation to RP but also that, withholding isotopologue overlap due to double bond differences or other ion forms, the semi-quantification of distinctly resolved GSL species could be readily achieved. This feature is observed for both negative and positive ion modes (Supplementary Fig. 3). This capability enables ascribing likely GSL classes to unknown components for *de-novo* structure identification and prospective development of prediction tools of CCS and RT for GSLs.

Noteworthy, beside subclasses present in the customized standard mix: monosialomonohexosylganglioside (GM4), GM3, GM2, GM1, disialodihexosylganglioside (GD3), disialotrihexosylganglioside (GD2), GD1a, GD1b, b-series trisialotetrahexosylganglioside (GT1b), and b-series tetrasialotetrahexosylganglioside (GQ1b), we also identified other species/subclasses previously not reported in publications or by the provider. These include: Fuc-GD1, GalNAc-GD1, *O*-acetyl GD1, Neu5Gc-containing GD1, and pentasialotetrahexosylganglioside (GP1) (Figs. 1c and 2a–c, Supplementary Data 1). These species, otherwise denoted as impurities, served a valuable purpose here: (a) to infer experimental MS/MS spectra of such species not readily identifiable in biological mixtures, (b) to determine their ion mobility and/or CCS values, and (c) assert the sensitivity and structural resolution capability of the RP-LC-TIMS-PASEF for GSLs. Similarly, a deeper coverage of structural heterogeneity than previously reported or, frankly expected, was obtained for the complex extract of GSLs from porcine brain, which constitutes a reference biological extract (Fig. 2d). Here, fucosylated GD1 and GM1 species, Neu5Gc-containing GD1, GD3, GM3, and GT1b, respectively, *O*-acetylated GD1, GD3, GQ1b, and GT1b were unambiguously identified (Fig. 2, Supplementary Fig. 4, Supplementary Table 2). We detected and characterized 47 individual gangliosides of 17 subclasses in porcine brain extract, e.g., the highest number of species/subclasses reported so far in this mixture[28,39].

These data attest that our 4D-RP-TIMS-PASEF glycosphingolipidomics is amenable for highly sensitive detection and high-resolution profiling of medium- and long-chain GSLs, highly sialylated, fucosylated, and *O*-acetylated GSLs.

### Generation of in-house 4D-glycosphingolipid library for automated annotation

The absence of LC-MS experimental fragmentation spectra in annotation databases presents a notable challenge for GSLs structural characterization. Sialylated GSLs share fragmentation characteristics of sialic acid moiety and glycosidic bond cleavages between monosaccharide units in the chain, which are used for manual structural elucidation and curation, and are expected to be the basis of the GSLs rule-based annotation module in lipidomic softwares[40]. Many classes and subclasses of GSLs, including GD, GT, GQ, GP, fucose-substituted GSLs, and *O*-Acetyl modified sialylated GSLs, are not currently covered even by *in-silico* fragmentation spectra, owing to a general lack of specialized expertise required to dissect the fragmentation patterns for highly sophisticated structural architectures of sialylated GSLs. Again, different levels of branching, linkage type and positions, formation of ring-cleavages across different monosaccharide units, the

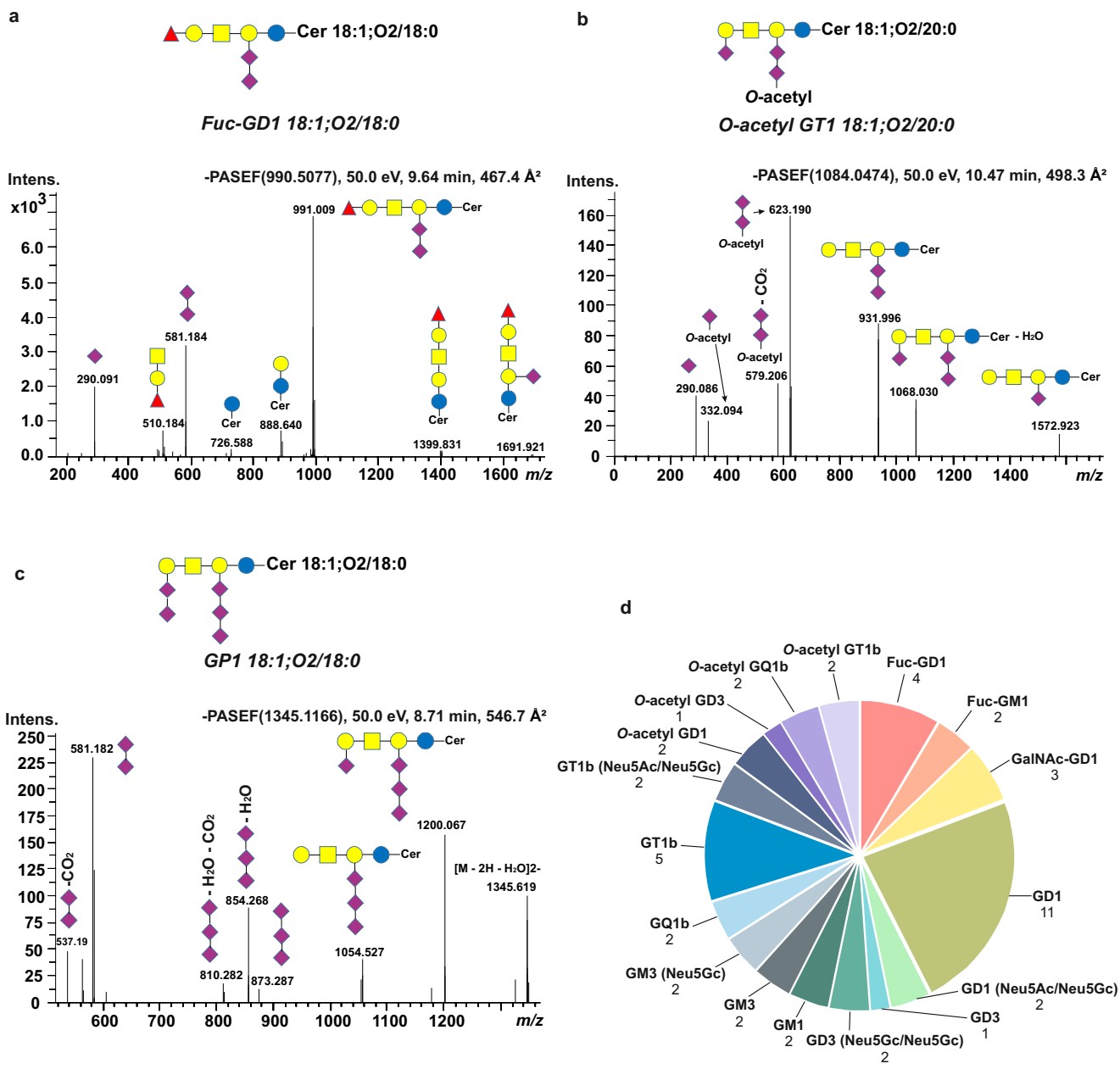

**Fig. 2 | Representative PASEF spectra for structural elucidation and determination of *O*-acetylated, fucosylated, and pentasialylated GSLs in standard mixtures and porcine brain extract. a** PASEF spectrum of Fuc-GD1 18:1;O2/18:0; **b** PASEF spectrum of *O*-acetyl GT1 18:1;O2/20:0; **c** PASEF spectrum of GP1 18:1;O2/18:0. Blue circle represents Glucose, yellow circle represents Galactose, yellow square represents *N*-acetylgalactosamine, purple diamond represents *N*- acetylneuraminic acid, and red triangle represents Fucose; **d** Identified gangliosides in 17 subclasses from porcine brain extract (Supplementary Table 1). Porcine brain extract was obtained from Avanti Polar Lipids. The extract was dissolved in MeOH/ water (8:2, v/v) to achieve a final concentration of 5 pmol/μL, with 20 μL injected onto the column. Source data are provided as a Source Data file.

effect of various substitutions like *O*- and *N*-acetylation, *N*-glycolyl, are a few of the factors challenging such a task. This hinders routine confident GSLs annotation and restricts the mid- to large-throughput analysis. Currently, the available lipid databases, such as MSDIAL, only comprise simple neutral GSLs, sulfatides, and GM3 species. These entries are further limited to their CCS and MS/MS spectra generated through in-silico simulation. Therefore, these data also lack the biological matrix and solvent effects for an effective and confident spectral matching-based annotation in biological matrices.

To overcome this limitation and assess whether GSLs extracted and enriched by fractionation (refer below to the extraction section)

can be at all detected and structurally characterized by RP-LC-TIMS-PASEF, we generated a first experimental 4D in-house database library including RT, CCS, *m/z*, and PASEF spectra. This database was compiled using commercially available GSL standards (Supplementary Table 1) and ganglioside extract from porcine brain. In total, we generated a library of 226 gangliosides from the referential porcine brain mix and standards (Supplementary Fig. 3 and Supplementary Fig. 4). This library- named *4D-sialoGSL standards library*- links the *m/z values* and PASEF spectra with RT and CCS values into the so-called Analyte List[32], and was subsequently used as basis to annotate the serological sialo-glycosphingolipidome.

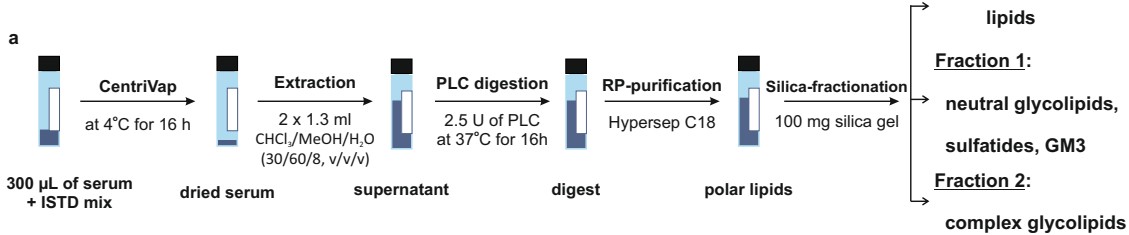

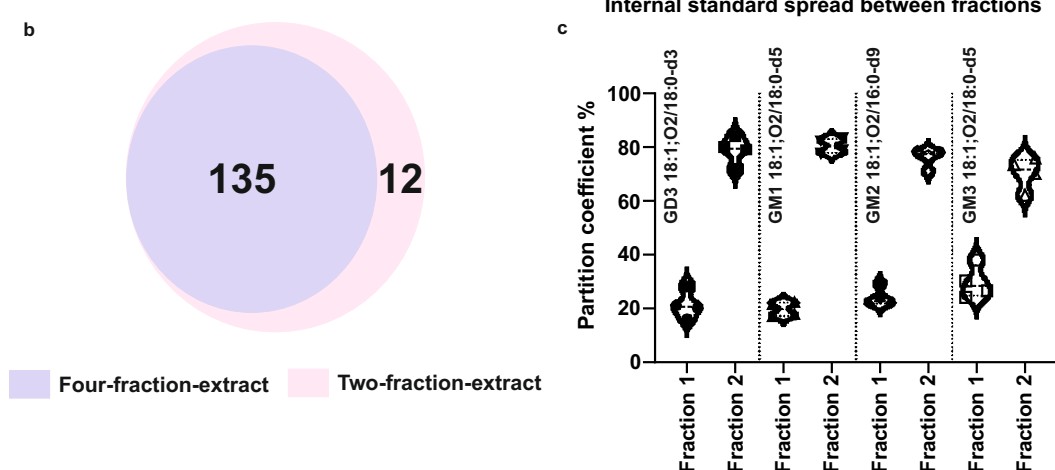

**Fig. 3 | Microextraction of glycosphingolipids in human serum. a** Schematic overview of serological glycosphingolipids (GSLs) extraction method; **b** Profiled coverage of sialylated GSLs in four-fraction extract and two-fraction extract (Supplementary Data 2 and 3); **c** Partition of internal standards calculated based on absolute intensity of each internal standard between two fractions in inter-day extract replicates of reference serum ($n = 4$). 300 μL of reference serum from volunteers was used for each extraction. Source data are provided as a Source Data file.

## Microextraction of glycosphingolipids in human serum

In a prior work, the extraction of mid- to long-chain sialylated GSLs from 300 μL of human serum was developed. However, for the subsequent identification of GSLs alteration with disease, specifically pancreatic and stomach cancer, sensitive analysis by nanoflow HPLC/nanoESI-QTOF MS was required. The long-chain sialylated species were primarily detected and structurally determined in pooled extracts, due to the overall sensitivity of the workflows achievable back then[12].

We leveraged this prior work[12] to optimize the GSLs extraction efficiency in order to match it with the μL-flow RP-LC separation efficiency and throughput and sensitivity of the 4D-TIMS-PASEF profiling of sialylated GSLs analysis described above. The extraction procedure comprises several key steps: (i) depletion of the phosphoglycerolipidome, (ii) RP purification, and (iii) silica-fractionation to enrich (sub)fractions of neutral and sialylated GSLs (Fig. 3a). The silica-fractionation step was here optimized to yield only two fractions, as opposed to the original four[12], resulting in substantial improvement of the sample processing, analysis and data processing time, respectively. It also reduces the variability of subsequent GSLs semi-quantification and the complexity of data processing (Fig. 3b). A higher number of complex GSLs were detected with the two-fraction versus four-fraction procedure, e.g., 147 (Supplementary Data 2) versus 135 GSLs (Supplementary Data 3), likely also due to reduced sample loss with the former one (Fig. 3b). Pooling fractions resulted in higher noise levels, increased matrix effects, and reduced the coverage of sialylated GSLs. Increasing serum volume to 450 μL did not improve coverage.

Another reason to optimize the silica-fractionation into only two fractions was to minimize the spread of internal standards (ISTDs) across fractions. For GSLs, deuterated ISTDs are poorly available, even less than for global lipidomics; they are not even available for some classes, i.e., GT, GQ, etc. Hence, the use of cross-class ISTDs is required. During the fractionation steps (Fig. 3a), the ISTDs have the propensity to partition in different fractions further challenging the semi-quantification. Four ISTDs: GD3 18:1;O2/18:0-d3, GM1 18:1;O2/18:0-d5, GM2 18:1;O2/16:0-d9, and GM3 18:1;O2/18:0-d5 were spiked into serum samples and the partition percentage between two fractions was assessed in replicates extracts of reference serum (Fig. 3c) in order to infer an optimal strategy for GSLs semi-quantification and evaluate its reproducibility and accuracy. The average partition of these ISTDs between the two fractions was found to be consistent and reproducible, indicating the effectiveness of the optimized extraction method. In fraction 2, the partition percentages of the ISTDs, GD3 18:1;O2/18:0-d3, GM1 18:1;O2/18:0-d5, GM2 18:1;O2/16:0-d9, and GM3 18:1;O2/18:0-d5 were: 78.70%, 80.40%, 75.99%, and 70.35%, with a coefficient of variation (CV) of 5.59%, 2.94%, 3.95%, and 7.40%, respectively. This consistent ISTDs partition was leveraged to semi-quantify GSLs in human serum (see below).

## Human serological glycosphingolipidome

As a first proof of concept of the analytical performance of combined GSL microextraction and 4D-glycosphingolipidomics, we conducted GSLs analysis in replicate reference serum samples, e.g., NIST serum 1951c. Of note, this serum was not officially certified for its GSL content. To expand our discovery of the glycosphingolipidome structural heterogeneity of human serum, we included serum samples of volunteers[32].

Our study demonstrates that the here developed serological GSLs extraction method (Fig. 3a), in conjunction with the μL-flow RP-LC-

TIMS-PASEF, outperforms the prior nanoflow LC-MS[12] in terms of structural heterogeneity coverage of sialylated GSLs (Fig. 3d, Supplementary Data 2–4). In the previous study, a total of 54 sialylated GSLs, including GSLs with up to 15 monosaccharides, such as sialylated difucosylated nLc12 species, were evidenced in human serum. The advancement obtained here on the structural profiling of GSLs is primarily attributed to improved sialylated GSLs coverage with two-step fractionation (Fig. 3b) and improved structural resolution by RP-LC-TIMS. Noteworthy in this context is the example of TIMS-separation of glycan-chain-based isomers of GM1 18:1;O2/18:0 and Neu5Ac-nLc4Cer 18:1;O2/18:0 from fraction 2, and their distinct fragmentation by PASEF (Fig. 4a) conducive to unambiguous structural elucidation. TIMS, but not RP-LC, separated the isobaric isomers GM1 and Neu5Ac-nLc4Cer (Fig. 4a), showcasing the orthogonal separation capability of TIMS to RP-LC. This complementary structural resolution by RP- LC and IMS enables a better dissection of the GSLs heterogeneity arising from ceramide diversity, glycan chain isomerism, composition, and length, and their subsequent structural determination by PASEF.

In fraction 2, 159 sialylated GSLs (Table 1, Supplementary Data 4) were detected and structurally elucidated from PASEF fragmentation patterns (Fig. 4a–c). Preservation of the high sialylation degree in intact GSL species, such as in GQ1b, that is without in-source fragmentation due to high flow rate and ionization energy, attests to good ionization efficiency to subsequently structurally characterize such species (Fig. 4b). The PASEF spectra in negative ion mode feature fragment ions arising from glycosidic bond cleavages, with preserved di-sialo or mono-sialo entities that are essential to dissect their location across the chain and the glycan chain configuration. This is well evidenced by the PASEF spectra of low-abundant GQ1b 18:1;O2/18:0 (Fig. 4b) and Neu5Ac-nLc6Cer 18:1;O2/14:0 (Fig. 4c).

The overall good sensitivity and structural resolution of the workflow is also supported by the detection of rare GM3 species containing short-chain fatty acyls, C10, C12, C14 (Table 1, Supplementary Data 4) primarily originating and detected in microbiota and dietary sources[41,42]. PASEF of these species in positive ion mode was performed to unambiguously confirm the fatty acyl content of the ceramide by the diagnostic fragment ions of sphingoid base and the corresponding neutral loss of fatty acyl from the ceramide ions (Supplementary Figs. 5 and 6). Similar as for standards (Supplementary Figs. 2–4; Supplementary Data 1) we valorized the positive ion mode fragmentation pattern for the sialylated GSLs in serum (generation of sphingoid base-diagnostic ions such as in Supplementary Figs. 5 and 6, Supplementary Data 4) to infer the sphingoid types contained in ceramide moieties and correspondingly the fatty acyl substitution (Supplementary Data 4).

Collectively, the extraction method for GSLs and the orthogonal capability of 4D-RP-LC-TIMS to dissect structural heterogeneity with subsequent procurement of PASEF-based structural data of individual species are conducive to extended coverage of the notoriously challenging sialylated glycosphingolipidome. Therefore, an enhanced coverage of GSLs exhibiting a higher degree of sialylation than detected before, e.g., tetrasialylated ganglioside, O-acetyl modification of sialic acid, and a higher heterogeneity of ceramides in GSLs was obtained. The total of 159 species covered 20 subclasses of sialylated GSLs in human serum (Table 1, Fig. 5, Supplementary Data 4), which is the highest number of sialylated GSLs of ganglio- and neolacto-series identified so far in human serum/plasma[12,17].

Moreover, 145 neutral GSLs in 6 subclasses and 72 sulfatides were identified in fraction 1 of the serum extract (Fig. 5, Supplementary Data 5 and 6, Supplementary Fig. 7). These GSLs feature N-linked fatty acyl chains ranging from C10 to C24, with up to 2 double bonds and up to 1 hydroxyl group, and varying sphingoid bases, predominantly 18:1;O2 (Supplementary Data 5 and 6). Again, PASEF fragmentation in positive ion produced sufficiently informative spectra to unequivocally derive the structure of both carbohydrate chain and ceramide

as exemplified in Fig. 4d for HexNAcHex3Cer d18:1;O2/16:0 and SHex2Cer d18:1;O2/16:0 (Fig. 4e).

Utilizing this 4D-glycosphingolipidomics workflow, we have successfully detected Fuc-GD1, GD2, GQ1b, and O-acetyl GD1 in human serum, demonstrating the method's sensitivity and capability. Additionally, we identified long-chain sialylated neolacto (nLc)-species ranging from nLc4 to nLc12 with varying degrees of fucosylation, and the broadest coverage of sulfatides in human serum so far. Although the fractionation procedure we employed was not geared for neutral GSLs[24] nor for sulfatides, the coverage of these subclasses is substantial and allows, in general, a more expanded investigation of GSL pathways with diseases. Noteworthy, the orthogonality of the RP-LC and IMS separation is well evidenced for all three main GSL classes: sialylated GSLs, neutral GSLs, and sulfatides, where the glycan chain extension and configuration are predominantly resolved by IMS, while ceramide composition is resolved by RP-LC (Fig. 5, Supplementary Fig. 7).

This deep coverage and precise structural profiling of sialylated GSLs, particularly at an overall throughput higher than before, holds great potential in deepening our understanding of their involvement in biological mechanisms and disease processes. Essentially, the parallel extraction and fractionation for 24 samples takes 2.5 days. An analytical throughput of 20 min per sample and about 1 h per batch for structural annotation results in an overall throughput of 24 samples/2.5 days for human serological GSLs profiling. We consider the 4D-glycosphingolipidomics platform−GSLs microextraction combined with 4D-RP-LC-TIMS-PASEF−to be a substantial technological advancement of serological GSL profiling in terms of coverage, robustness, sensitivity, and throughput.

We have created also a first version of comprehensive 4D-TIMS-PASEF experimental spectral library of glycosphingolipidome (MassIVE repository [https://doi.org/10.25345/C53J39C93], Research Tools [https://www.unimedizin-mainz.de/lipidomics-unit/lipid-research/research-tools.html], github [https://github.com/dtots17/Lipidomics-Spectral-Database.git], and Zenodo [https://doi.org/10.5281/zenodo.14882123]) that will enable the researchers a more streamlined annotation and curation of GSLs in samples of various origins. This resource is expected to expedite the annotation and investigation of GSLs' spatial biology by ref. 43.

The 159 sialylated GSLs, 145 neutral GSLs, and 72 sulfatides found in human serum were used to generate three comprehensive 4D-libraries, called *4D-sialylated GSL serum library, 4D-neutral GSLs serum library*, and *4D-sulfatides serum library* applied subsequently for routine and confident annotation of serological GSLs.

## Semi-quantification strategies for sialylated glycosphingolipids in human serum

Relative quantification of sialylated GSLs obtained from human serum by extraction and fractionation into two fractions is challenged not only by the lack of ISTDs even for each class and their partition across the fractions, but also by data processing. For the latter, examples include: (i) peak picking of singly and multiply charged ions of GSLs, which exhibit different isotopologue distribution than that of peptides for which most of the processing softwares are developed for, require manual curation to avoid faulty inferring of peak areas and quantification; (ii) semi-quantification of isomers resolved only in ion mobility dimension; (iii) algorithm performance for extraction of features with near-overlapping chromatographic separation, (iv) relative quantification of fully overlapping isomers, in RT and CCS dimensions, but distinguishable by one or more fragment ions, etc. The points highlighted at (i)–(iii) were here addressed by manual curation, while the latter point will be subject to further developments of MS/MS-based quantification and/or other chromatographic methods coupled with TIMS-PASEF. To achieve a reproducible semi-quantification of sialylated GSLs in individual samples, rather than in pooled sample extracts

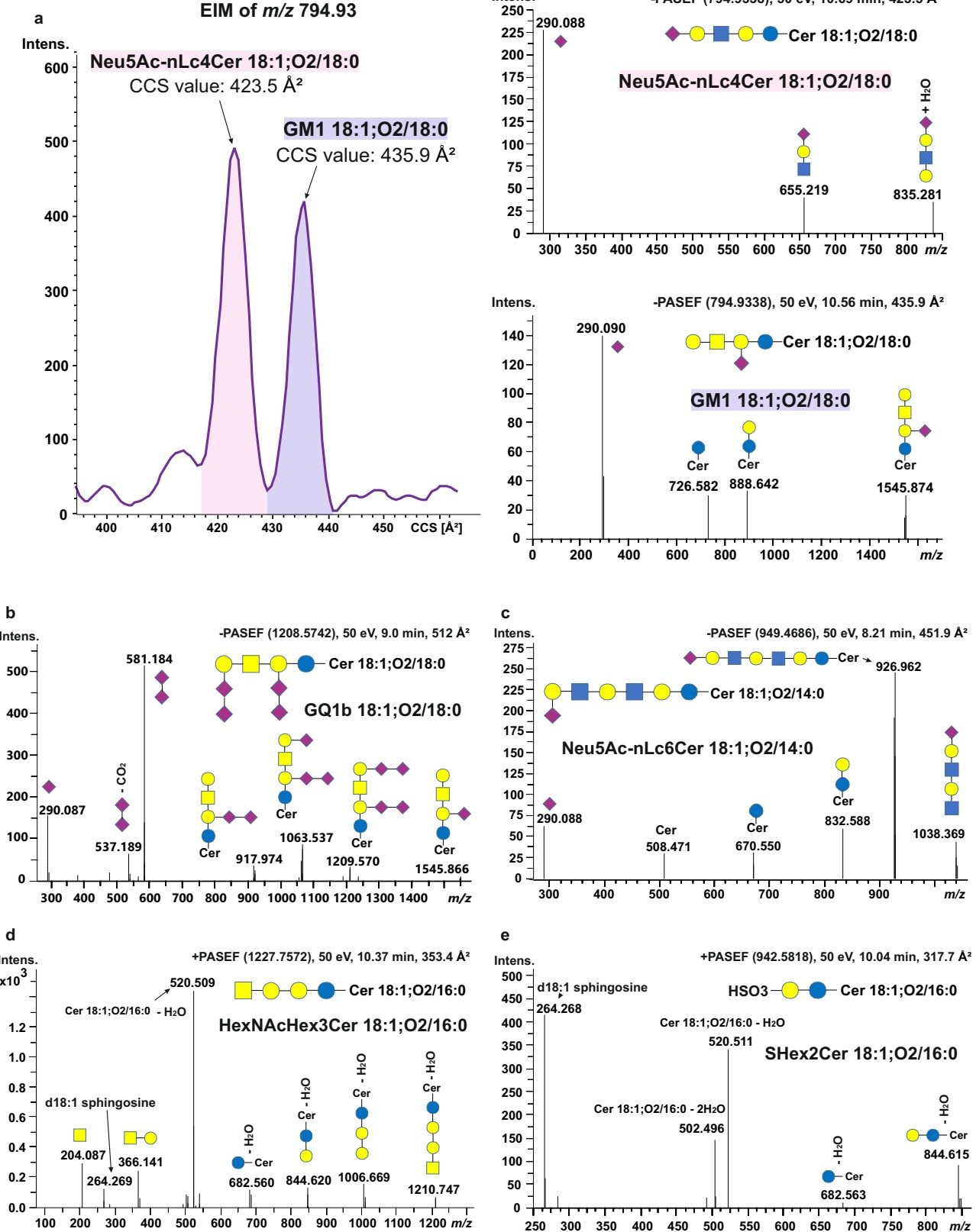

**Fig. 4 | Representative PASEF spectra of glycosphingolipids in human serum.**
**a** Baseline separation of Neu5Ac-nLc4Cer 18:1;O2/18:0 and GM1 18:1;O2/18:0 in extracted ion mobilogram at *m/z* 794.93; **b** PASEF spectra of GQ1b 18:1;O2/18:0; **c** PASEF spectra of Neu5Ac-nLc6Cer 18:1;O2/14:0; **d** PASEF spectra of HexNAcHex3Cer 18:1;O2/16:0; **e** PASEF spectra of SHex2Cer 18:1;O2/16:0.

PASEF spectra were acquired using a collision energy of 50 eV. Blue circle represents Glucose, yellow circle represents Galactose, yellow square represents *N*-acetylgalactosamine, purple diamond represents *N*-acetylneuraminic acid, and red triangle represents Fucose.

**Table 1 | Glycosphingolipid species detected by µL-flow 4D-RP-LC-TIMS-PASEF analysis in human serum**

| Class | Subclass | Sphingoid base | Fatty acyl chain length | Number of double bonds in the fatty acyl chain | Degree of hydroxylation in fatty acyl chain | Number of species | Total |
|---|---|---|---|---|---|---|---|
| Gangliosides | GM3 | 18:0;O2 | C14–C22 | 0 | 0 | 5 | 66 |
| | | 18:1;O2 | C10–C26 | 0–3 | 0 | 37 | |
| | | 18:0;O3 | C16–C22 | 0 | 0 | 2 | |
| | | 18:1;O3 | C10–C24 | 0–2 | 0 | 22 | |
| | GM2 | 18:1;O2 | C16–C18 | 0–1 | 0 | 3 | 3 |
| | GM1 | 18:1;O2 | C16–C18 | 0 | 0 | 2 | 2 |
| | GD3 | 18:0;O2 | C16 | 0 | 0 | 1 | 17 |
| | | 18:1;O2 | C14–C24 | 0–2 | 0 | 15 | |
| | | 18:1;O3 | C16 | 0 | 0 | 1 | |
| | GD2 | 18:1;O2 | C16–C24 | 0–1 | 0 | 4 | 4 |
| | GD1 | 18:0;O2 | C16 | 0 | 0 | 1 | 18 |
| | | 18:1;O2 | C14–C24 | 0–2 | 0 | 15 | |
| | | 18:1;O3 | C16–C18 | 0 | 0 | 2 | |
| | GT1b | 18:0;O2 | C16 | 0 | 0 | 1 | 13 |
| | | 18:1;O2 | C14–C24 | 0–2 | 0 | 12 | |
| | GQ1b | 18:1;O2 | C16–C18 | 0 | 0 | 2 | 2 |
| | O-acetyl GD1 | 18:1;O2 | C16–C18 | 0–1 | 0 | 3 | 3 |
| | Fuc-GD1 | 18:1;O2 | C20 | 0 | 0 | 1 | 1 |
| Sialylated neolacto-series | Neu5Ac-nLc4Cer | 18:1;O2 | C14–C24 | 0–2 | 0 | 10 | 10 |
| | Neu5Ac-nLc6Cer | 18:1;O2 | C16–C24 | 0–2 | 0 | 6 | 6 |
| | Neu5Ac-nLc8Cer | 18:1;O2 | C16–C24 | 0–1 | 0 | 3 | 3 |
| | Neu5Ac-nLc10Cer | 18:1;O2 | C16 | 0 | 0 | 1 | 1 |
| | Neu5Ac-Fuc-nLc6Cer | 18:1;O2 | C16–C18 | 0–1 | 0 | 3 | 3 |
| | Neu5Ac-Fuc-nLc8Cer | 18:1;O2 | C16–C24 | 0–1 | 0 | 3 | 3 |
| | Neu5Ac-Fuc-nLc10Cer | 18:1;O2 | C16 | 0 | 0 | 1 | 1 |
| | Neu5Ac-Fuc-nLc12Cer | 18:1;O2 | C16 | 0 | 0 | 1 | 1 |
| | Neu5Ac-Fuc2-nLc10Cer | 18:1;O2 | C16 | 0 | 0 | 1 | 1 |
| | Neu5Ac-Fuc2-nLc12Cer | 18:1;O2 | C16 | 0 | 0 | 1 | 1 |
| Neutral glycosphingolipids | HexCer | 16:1;O2 | C18–C24 | 0–1 | 0–1 | 8 | 45 |
| | | 17:1;O2 | C24 | 0 | 0 | 1 | |
| | | 18:1;O2 | C16–C24 | 0–2 | 0–1 | 19 | |
| | | 18:2;O2 | C16–C24 | 0–1 | 0–1 | 13 | |
| | | 18:0;O3 | C16–C24 | 0–1 | 0–1 | 3 | |
| | | 18:0;O4 | C24 | 1 | 0 | 1 | |
| | Hex2Cer | 16:1;O2 | C14–C24 | 0–2 | 0–1 | 11 | 53 |
| | | 16:2;O2 | C16 | 0 | 0 | 1 | |
| | | 17:1;O2 | C16–C24 | 0–1 | 0 | 4 | |
| | | 17:2;O2 | C16 | 0 | 0 | 1 | |
| | | 18:0;O2 | C16 | 0 | 0 | 1 | |
| | | 18:1;O2 | C12–C24 | 0–2 | 0–1 | 19 | |
| | | 18:2;O2 | C12–C24 | 0–2 | 0–1 | 14 | |
| | | 18:1;O3 | C16 | 0 | 0 | 1 | |
| | | Unidentified sphingoid base | Unidentified fatty acyl chain | – | – | 1 | |
| | Hex3Cer | 16:1;O2 | C12–C22 | 0 | 0–1 | 6 | 34 |
| | | 17:1;O2 | C16–C18 | 0 | 0 | 2 | |
| | | 18:0;O2 | C16 | 0 | 0 | 1 | |
| | | 18:1;O2 | C10–C24 | 0–2 | 0–1 | 13 | |

**Table 1 (continued) | Glycosphingolipid species detected by μL-flow 4D-RP-LC-TIMS-PASEF analysis in human serum**

| Class | Subclass | Sphingoid base | Fatty acyl chain length | Number of double bonds in the fatty acyl chain | Degree of hydroxylation in fatty acyl chain | Number of species | Total |
|---|---|---|---|---|---|---|---|
| | | 18:2;O2 | C12–C24 | 0–1 | 0–1 | 11 | |
| | | Unidentified sphingoid base | Unidentified fatty acyl chain | – | – | 1 | |
| | HexNAcHex3Cer | 16:1;O2 | C16 | 0 | 0 | 1 | 11 |
| | | 17:1;O2 | C16 | 0 | 0 | 1 | |
| | | 18:1;O2 | C14–C24 | 0–1 | 0 | 5 | |
| | | 18:2;O2 | C16–C22 | 0 | 0 | 2 | |
| | | Unidentified sphingoid base | Unidentified fatty acyl chain | – | – | 2 | |
| | HexNAcHex4Cer | 18:1;O2 | C16 | 0 | 0 | 1 | 1 |
| | Fuc-HexNAcHex3Cer | 18:1;O2 | C16 | 0 | 0 | 1 | 1 |
| Sulfatides | SHexCer | 16:1;O2 | C16–C24 | 0–1 | 0–1 | 7 | 58 |
| | | 17:1;O2 | C16 | 0 | 0–1 | 2 | |
| | | 18:0;O2 | C16 | 0 | 0–1 | 2 | |
| | | 18:1;O2 | C14–C24 | 0–1 | 0–1 | 17 | |
| | | 18:2;O2 | C16–C24 | 0–1 | 0–1 | 14 | |
| | | 16:0;O3 | C18–C24 | 0–1 | 0–1 | 3 | |
| | | 18:0;O3 | C16–C24 | 0–1 | 0–1 | 5 | |
| | | 18:1;O3 | C16 | 0 | 1 | 1 | |
| | | 18:0;O4 | C24 | 2 | 0 | 1 | |
| | | Unidentified sphingoid base | Unidentified fatty acyl chain | – | – | 6 | |
| | SHex2Cer | 16:1;O2 | C18 | 0 | 1 | 1 | 14 |
| | | 18:1;O2 | C16–C24 | 0–1 | 0–1 | 5 | |
| | | 18:2;O2 | C16–C24 | 0–1 | 0 | 4 | |
| | | Unidentified sphingoid base | Unidentified fatty acyl chain | – | – | 4 | |

as previously possible[12], we set out to devise strategies to account for ISTD partition across the two extraction fractions and cross-class ISTDs performance.

Two strategies were first evaluated to account for ISTD partition between the two fractions and enable eventual calculation of GSLs concentration per μL serum, for an easier read-out and comparison across samples and studies (Fig. 6a).

Strategy (I): This strategy makes use of control serum extracts (i.e., from pooled serum). The average partition coefficient of each ISTDs in the two fractions of $n = 6$ control serum extracts was used to calculate and revert the levels per μL serum of sialylated GSLs in each of the individual samples. To devise the average percentage partition of the ISTDs between two fractions (see also Fig. 3c), we semi-quantified the GSLs in six inter-day serum extractions of an individual volunteer (refer to the section on GSLs microextraction).

Strategy (II): The partition coefficient of ISTDs in the two fractions of each individual sample was used to calculate and revert the levels of sialylated GSLs in the respective individual samples. This strategy requires the analysis of both fractions for each sample.

The two strategies were also evaluated and compared in conjunction with different combinations of ISTDs per GSL class, referred here as sub-strategies (Fig. 6b) to also account for cross-class ISTD performance.

The combinatorial use of the Strategy I and II, with Strategy II applied only to correct outlier sample cohorts, was also evaluated (Fig. 6a). The aim here was to devise a protocol for sialylated GSLs quantification that solely requires analysis of fraction 2, saving time for acquisition and processing data. In this case, fraction 1 is measured only to correct for outliers with Strategy I and/or if neutral GSLs and sulfatides are of interest for investigation.

To evaluate the performance of Strategies I and II, and combination thereof, we used a more permissive cut-off of 35% for the CV, given the low abundance of GSLs in human serum and poor availability of ISTDs. For Strategy I, the initial results showed rather low percentages of species with CVs less than 35%: 36.89%, 23.77%, and 41.80% for the first, second, and third sub-strategies, respectively. These low percentages were attributed to anomalous ISTD partition percentages (Supplementary Data 7) exhibited by two replicates. Excluding one outlier out of six inter-day replicates, the percentages of species with CVs less than 35% increased to 67.21%, 44.26%, and 76.23% for the three sub-strategies, respectively. When 2 outlier replicates were excluded, the percentages further improved to 87.80%, 84.42%, and 86.89% for the three sub-strategies, respectively (Fig. 6b).

If the partition of ISTDs in each sample was factored in as in the Strategy II, the percentages of species with CVs less than 35% across the six samples were 66.39%, 61.48%, and 70.49% for the three sub-strategies, respectively, without removing any outliers. As shown in Fig. 6b, this method of considering the partition of ISTDs in each sample yielded overall better results compared to quantifying based on an average percentage partition of ISTDs between two fractions. However, this approach requires analysis of both fractions instead of only fraction 2, resulting in increased time for data acquisition and processing if only sialylated GSLs are of interest.

Among the various combinations of ISTDs per GSL class, the third sub-strategy (Fig. 6b) showed the most comparable and reproducible results, e.g., less outliers, indicating that the analytes and their corresponding ISTDs behave similarly in extraction and measurement. Therefore, we propose adopting this sub-strategy in conjunction with Strategy II. If outlier samples/measurements are observed, the partition of ISTDs in each sample (Strategy I) can be used to recalculate and

# 376 glycosphingolipids in human serum

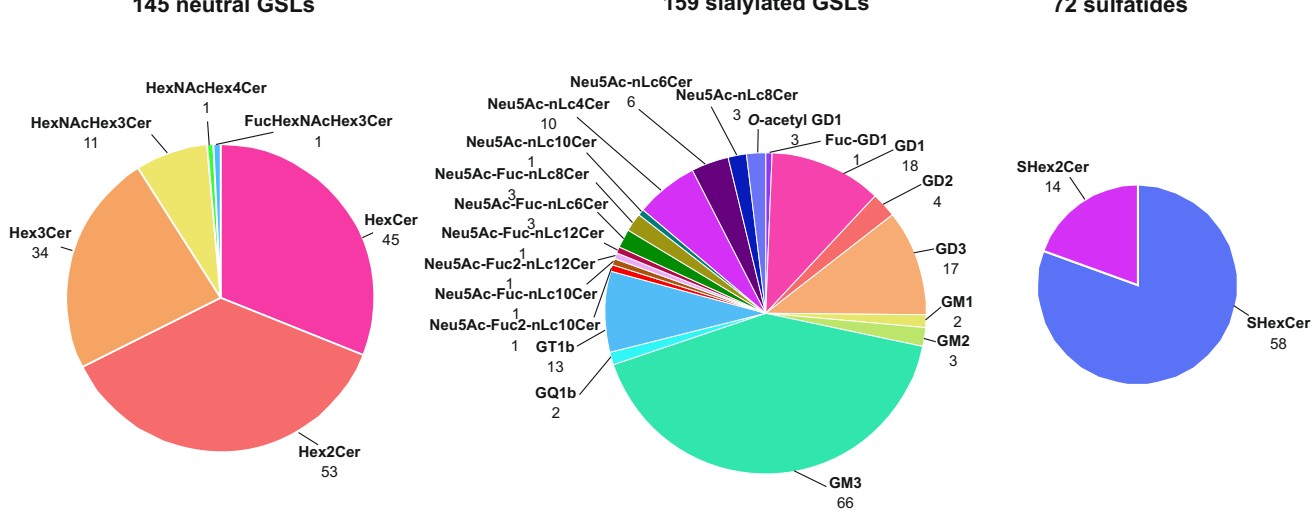

**145 neutral GSLs**

**Sphingoid base:** 16:1;O2, 16:2;O2, 17:1;O2, 17:2;O2, 18:0;O2, 18:1;O2, 18:2;O2, 18:0;O3, and 18:0;O4

**N-linked fatty acyl chain:** C10 to C24, number of double bonds: from 0 to 2, degree of hydroxylation: from 0 to 1

**159 sialylated GSLs**

**Sphingoid base:** 16:1;O2, 18:0;O2, 18:1;O2, 18:2;O2, 18:0;O3, and 18:1;O3

**N-linked fatty acyl chain:** C10 to C26, number of double bonds: from 0 to 3

**72 sulfatides**

**Sphingoid base:** 16:1;O2, 16:0;O3, 17:1;O2, 18:0;O2, 18:1;O2, 18:2;O2, 18:0;O3, 18:1;O3, and 18:0;O4

**N-linked fatty acyl chain:** C14 to C24, number of double bonds: from 0 to 2, degree of hydroxylation: from 0 to 1

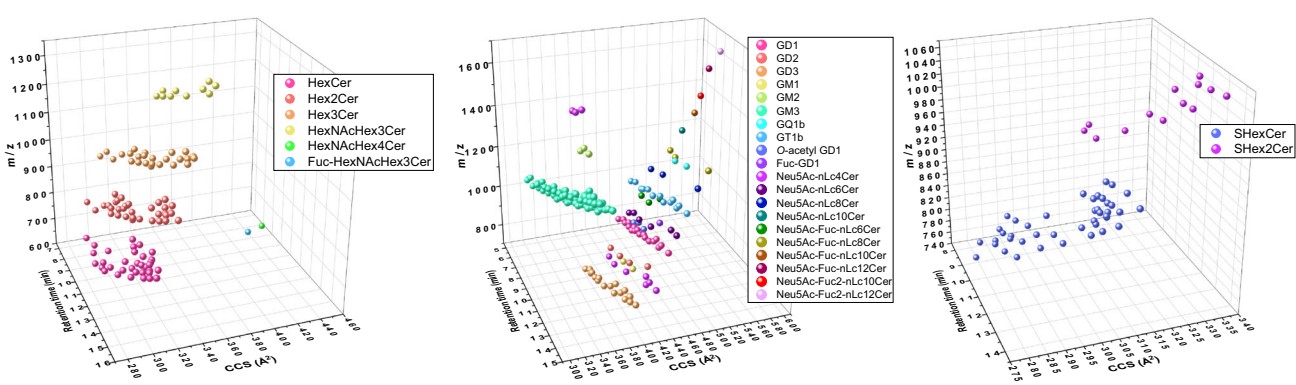

**Fig. 5 | Human serological glycosphingolipidome identified using µL-flow 4D-RP-LC-TIMS-PASEF.** The structural heterogeneity of the neutral GSLs, sialylated GSLs and sulfatides determined in the human serum is depicted in the pie charts. Overview of the sphingoid bases and N-linked fatty acyl chains occurring in human serum glycosphingolipidome is presented for each GSL class (middle panel). The efficient separation of GSLs obtained by combination of the RP-LC and ion mobility is well visible in the 3D space (m/z, RT, CCS) of each GSL class: neutral GSLs, sialylated GSLs, and sulfatides (lower panel). (Supplementary Data 4–6). Source data are provided as a Source Data file.

correct these samples. After correcting for two outliers, the percentage of species that exhibit CV below 35% improved to 72.95%. The species with CVs higher than 35% were either of very low abundance or largely influenced by factors such as sample preparation, extraction efficiency, or analyte stability. The detailed plot in Fig. 6c of concentration of sialylated GSLs determined in fraction 2 of replicate sera extracts and the average CV% per class shows that the CV for the majority of GSL classes lies below 30%.

Linearity parameters for all the major sialylated GSLs classes are all above 0.97; the lower limit of detection (LLOD) in the low fmol range, e.g., higher than 0.135 (for GQ1b), and lower limit of quantification (LLOQ) lies in the high fmol range, e.g., above 0.410 (for GQ1b). GD1 has the highest LLOQ of 2.2 pmol/µL, probably due to its GD1a and GD1b isomerism (Fig. 6d). The average carry-over effect across all was 1.2% (Supplementary Tables 3 and 4). For semi-quantification of neutral GSLs and/or sulfatides in fraction 1 in clinical samples, we

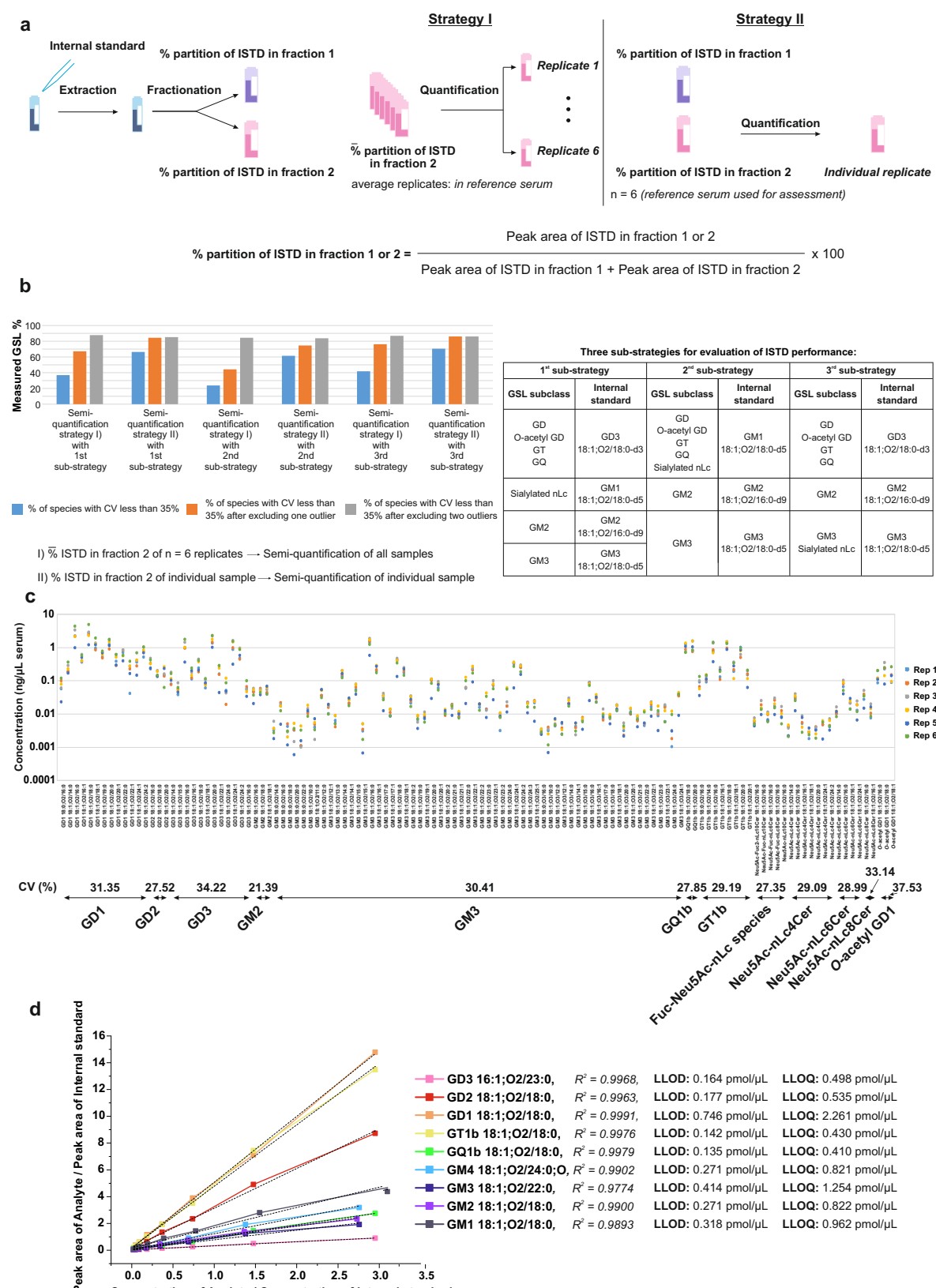

normalized to the peak area of the level-3 ISTD, GM3 18:1;O2/18:0 (see below).

## Expression of glycosphingolipids in Parkinson's disease

PD's hallmark is the aggregation of presynaptic α-synuclein, along with lipids and other cellular matter in the brain, and it has been reported that GM3 stimulates this aggregation[44]. Furthermore, mutation of the GBA1 gene, encoding the lysosomal enzyme glucocerebrosidase (GCase), is widely recognized as a risk factor for PD[45]. GCase catalyzes the hydrolysis of glucocerebroside into glucose and ceramide. This is an important step in the metabolic breakdown of GSLs. Reduced GCase activity has been observed in individuals with PD[46]. There have

**Fig. 6 | Quantification strategies for sialylated glycosphingolipids in human serum. a** Schematic workflow of two quantification strategies (I) and (II) for sialylated glycosphingolipids (GSLs) in human serum; **b** Comparison between quantification strategies. The bar graphs display the percentage of species with a coefficient of variation below 35%. This data is derived from quantifying six inter-day extraction replicates, with one measurement per serum extract (Supplementary Data 7); **c** Concentration of 122 sialylated GSLs in ng/μL serum plotted against sialylated GSL species identified from the volunteer's serum extracted and measured on two different days, with three replicates per day. This figure demonstrates the reproducibility of the semi-quantification of sialylated GSLs in human serum using the proposed combination of Strategy I and II, where the latter was used for correction in cases where outliers were detected; **d** Linearity curve representing the relationship between the concentration ratio (concentration of analyte/concentration of internal standard) on the *x*-axis and the response ratio (peak area of analyte/peak area of internal standard) on the *y*-axis of nine analytes. The figure includes the lower limit of detection (LLOD) and lower limit of quantification (LLOQ) for each analyte using its corresponding internal standard as 3rd sub-strategy. This curve was used to assess the analytical range and sensitivity for quantifying each analyte. Source data are provided as a Source Data file.

been reports of changes in HexCer levels in the blood of PD patients[47], but alterations in downstream biosynthetic products have not been documented.

To date, only a limited number of publications have documented alterations in GSLs in the serum of PD[48]. These reports mainly focused on quantifying subclasses rather than providing a comprehensive analysis of individual GSL species, due to prior analytical limitations.

Here, we applied our developed 4D-glycosphingolipidomics platform to profile and semi-quantify sialylated GSLs in human serum samples of 30 control (HC) and 28 PD patients to: (i) demonstrate the potential of this methodology to uncover PD-associated sialylated GSLs that can serve as a basis to discover specific biomarkers for PD and (ii) unravel disturbances in molecular processing associated with PD.

We identified 98 sialylated GSLs in sera from controls and PD patients, including 10 sialylated neolacto-species with up to 12 monosaccharides—in sialylated Fuc-nLc10—and up to trisialo ganglio-species. The ceramide moiety varied between Cer 18:1;O2/10:0 to Cer 18:1;O2/24:0. Additionally, 145 neutral GSLs, including Hex-, Hex2-, Hex3-, and HexNAcHex3Cer species, along with 72 sulfatides, were identified. These GSLs featured *N*-linked fatty acyl chains ranging from C10 to C24, with 0 to 2 double bonds and 0 to 1 hydroxyl group (Supplementary Data 8 and 9).

Statistical analysis using the Limma method with sex and age as covariates evidenced that PD is associated with 41 upregulated GSLs, out of which 39 gangliosides and 2 sialylated neolacto-species (Fig. 7a, b). Sex was found to be a major contributor to PD, while age was a rather random parameter (Supplementary Data 10).

From the ganglio-series, GM3, GM2, GD3, and GD1 exhibit a statistically significant increase in PD compared to controls. Sialylated nLc4 and nLc6 were also significantly increased in PD. From the biosynthetic pathways underscored by these significant changes (Fig. 7b) and by inferring corresponding data from the KEGG pathway (www.genome.jp) and reactions analysis (www.lipidmaps.org), it is deducible that the ST3GAL5, ST8SIA1, B4GALNT1, B3GALT4, ST3GAL1, B4GALT1-4, and GCNT2 enzymatic pathways are significantly increased with PD.

Noteworthy, sialylated GSLs exhibited a differential quantitative pattern between females and males in PD (Fig. 8a, b, Supplementary Data 11). GT1b 18:1;O2/20:0, GM3 18:1;O2/18:1, GM3 18:1;O2/22:2, Neu5Ac-nLc4Cer 18:1;O2/14:0, Neu5Ac-nLc4Cer 18:1;O2/16:0, and Neu5Ac-nLc4Cer 18:1;O2/16:1 were significantly increased in females compared to males, while GD1 18:1;O2/14:0, GD1 18:1;O2/15:0, GD1 18:1;O2/18:1, GD1 18:1;O2/20:0, GD1 18:1;O2/20:1, GD 18:1;O2/24:1, and GM3 18:1;O3/15:0 were significantly increased in males compared to females. Interestingly, the level of *O*-acetyl GD1 18:1;O2/16:0 was oppositely altered in females and males. Comparing changes with PD of GSL classes rather than at molecular species level, evidences a significantly increased GM2 and Neu5Ac-nLc4Cer in females, only suggesting a systemic alteration of the glycosyltransferase/glycosidase responsible for metabolism of these classes independent of ceramide metabolism.

These results corroborate well with the clinical data on the differential susceptibility of females and males to developing PD and the differential progression course of PD for males and females[49]. Hence, it intrinsically validates the method to render a reliable clinical phenotype read-out. No change was observed for the class of HexCer, which concurs with the fact that these patients were not diagnosed with the GBA1 mutation. Downstream extended neutral GSLs, Hex2Cer, Hex3-Cer, and HexNacHex3Cer are, however, significantly increased with PD (Fig. 8c, d, Supplementary Data 11). Such changes corroborate with those evidenced in sialylated GSLs, in that these species are substrates for glycosyltransferases and sialidases that actually synthesize the ganglio and neolacto-species upregulated in PD (Fig. 7b). The sulfatide subclasses remain unchanged with PD (Fig. 8c); however, SHexCer 32:1;O3 decreases, while SHexCer 40:1;O3 increases in PD (Fig. 8d).

## Discussion
### Method's utility and capability
The 4D-glycosphingolipidomics platform exhibits a higher throughput capability than previously reported, with complete processing of 24 samples in 2.5 days. It allows for extended coverage of sialylated GSLs in serum: 129 of sialylated GSLs cover ganglio-series, and 30 species cover neolacto-series. This is the highest coverage so far, due to our knowledge of the sialylated glycosphingolipidome in human serum.

Given that in reference to porcine brain extracts and commercially available standards, low-abundant pentasialo ganglioside, GP, GalNAc-GD1, and Neu5Gc-containing species were detected, it is fair to assert that the 4D-glycosphingolipidomics platform exhibits, as yet, an unparalleled sensitivity, particularly for a platform operating at μL-flow rate. This performance is further supported by the fact that sialic acid, otherwise readily fragile under ionization and ion transfer conditions, has been here preserved onto penta- and tetrasialo ganglio-species.

These results provide more insights into the biosynthetic sialylated GSLs machinery and/or secretion of GSLs in serum than unraveled with the coverage achieved by nanoflow LC-ESI from the same serum volume[12]. By combining orthogonal RP-LC and IMS, it allows for glycan chain-based separation, typically achieved by HILIC/normal phase chromatography, and with the added benefit of ceramide-based separation. The orthogonality of the separation afforded by the combination of RP-LC and TIMS is notably independent of GSL type: sialylated, neutral, and sulfatides (Fig. 5) and enhances the overall glycosphingolipidome coverage at the molecular species level without derivatisation or GSL structure modification by enzymatic approaches (ceramidases, sialidase, etc).

Our data reveal that serological sialylated glycosphingolipidome is much more complex than previously understood, i.e., contains: (i) extended chains of sialylated neolacto-series with up to 15 monosaccharide units for sialylated and bi-fucosylated nLc12, (ii) up to tetrasialo gangliosides, (iii) *O*-acetylated gangliosides, and (iv) each of the above exhibiting a diversity of fatty acyls and sphingoids in ceramide moieties. Seemingly, modifications of sialic acid by *O*-acetylation are not as uncommon for serological ganglio-species, since it is present on GD, GT, and GQ species, respectively. In total, we identified 159 individual sialylated GSL species covering 20 subclasses in human serum. Although fractionation of serum is necessary to enrich the sialylated glycosphingolipidome and expand its structural coverage, fraction 1 is also a very rich source of biological information. It contains 145 neutral

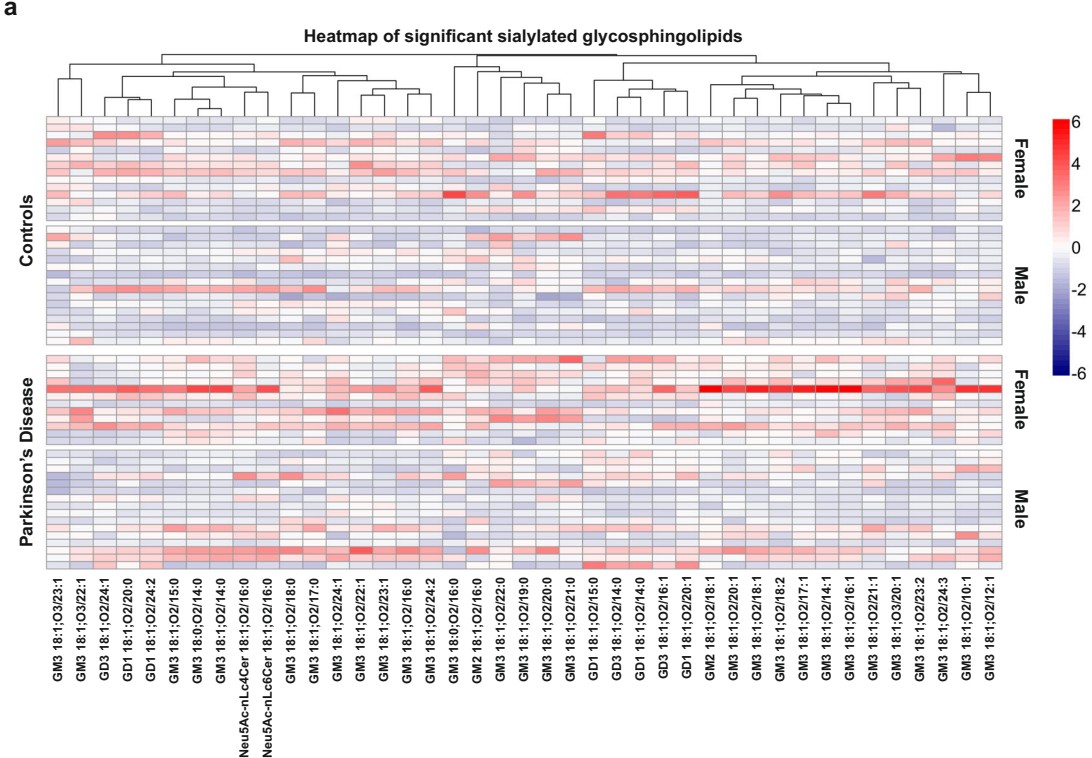

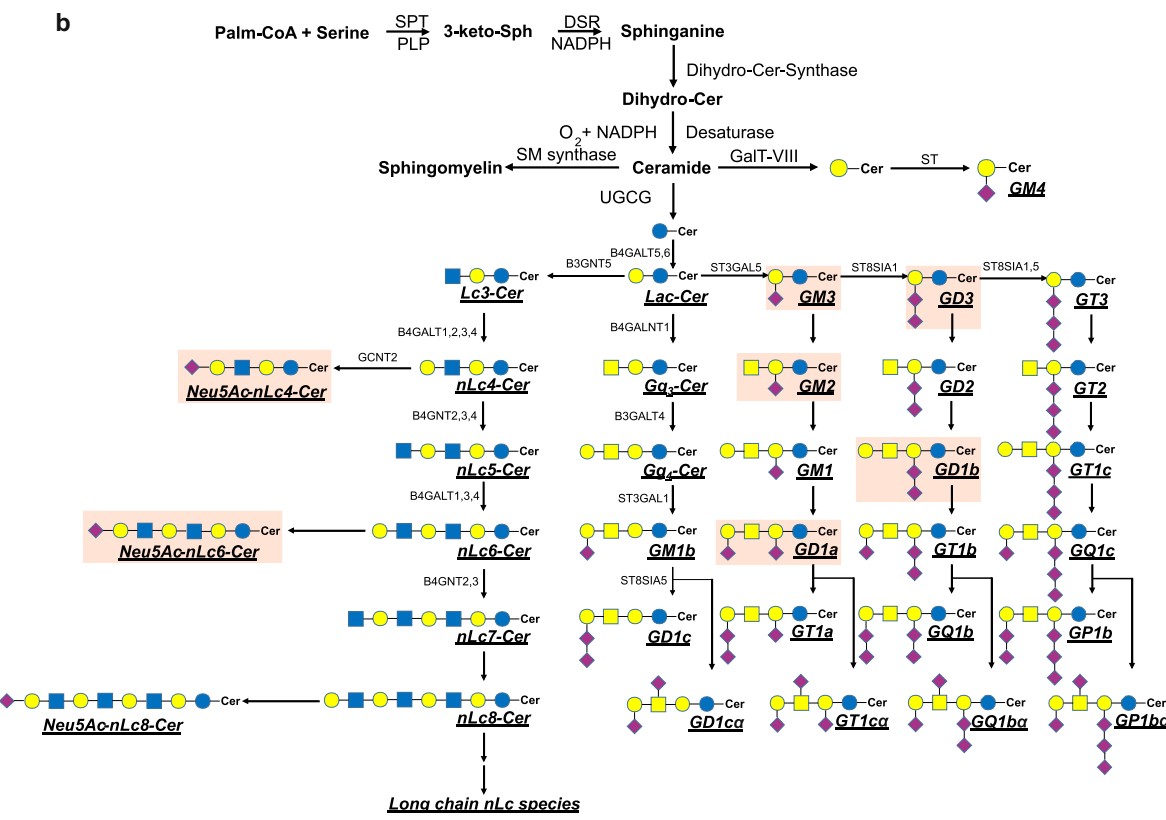

**Fig. 7 | Expression of sialylated glycosphingolipids in Parkinson's disease.**
**a** Heatmap showing the upregulated (top) and down-regulated (bottom) differentially expressed sialylated GSLs after FDR correction. For visualization purpose, raw expression values were organized according to group (control and Parkinson's Disease) and sex (female and male); **b** Biosynthetic pathway of ganglio- and neolacto-series (www.genome.jp and www.lipidmaps.org) of glycosphingolipids (GSLs) with highlighted upregulated subclasses. Source data are provided as a Source Data file.

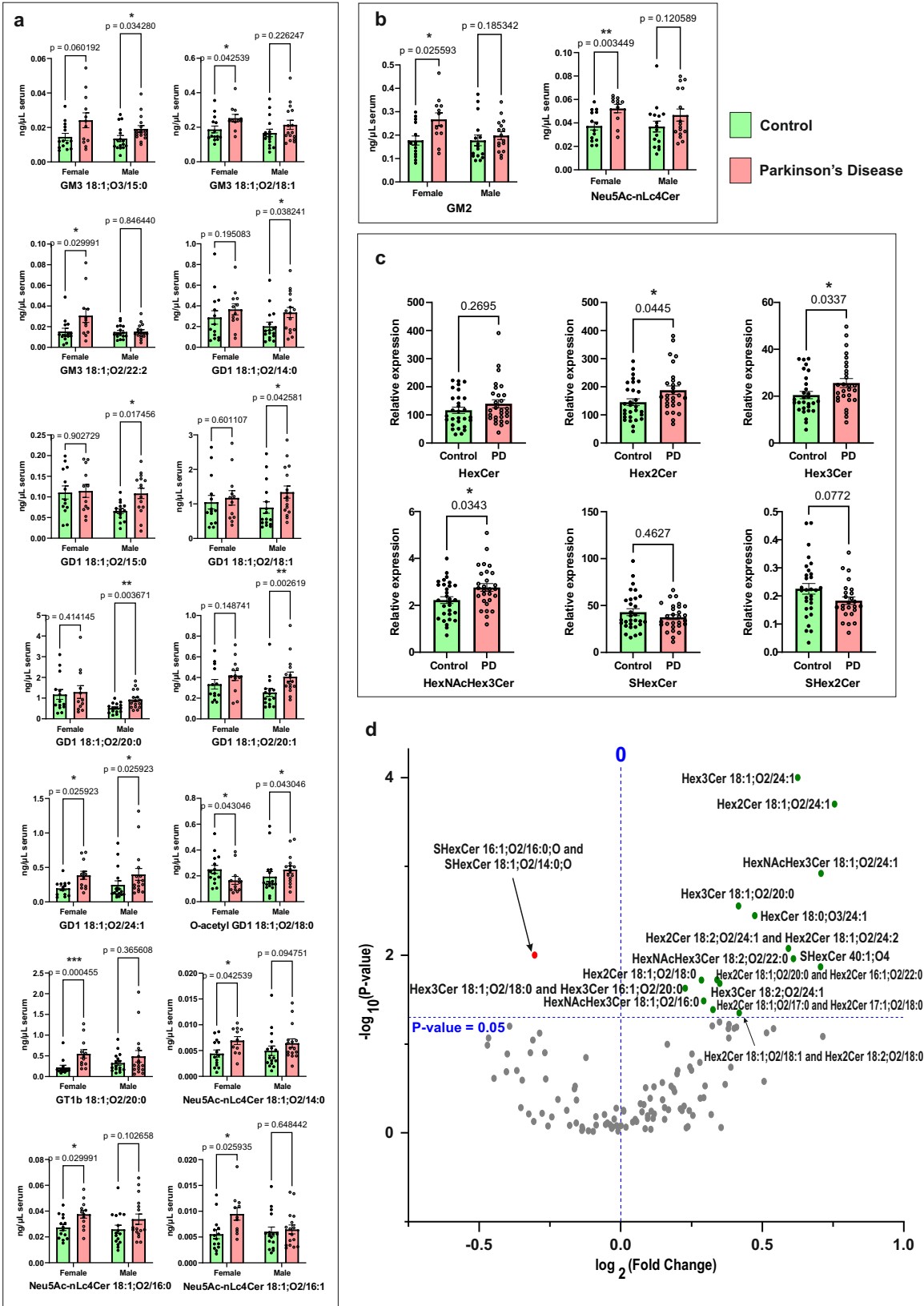

GSLs, covering classes, from Hex-to-(Fuc)HexNAcHex4Cer, which also serve, among other functions, as substrates for ganglio- and neolacto-series biosynthesis. Noteworthy, this fraction also contains 72 sulfatides of SHexCer and SHex2Cer, which is the largest number detected so far in human serum.

The excitingly diverse serological GSL pattern, of 376 total GSLs identified, prompts further investigation into the activity of glycosyltransferases and glycosidases involved in the biosynthesis of GSLs, particularly in relation to certain diseases, as well as the sources of these molecules in the human serum. Sialylated GSL series exhibit

**Fig. 8 | Differential quantitative pattern between control and Parkinson's disease. a** Non-parametric, two-sided multiple Mann–Whitney *U*-test results comparing individual sialylated glycosphingolipids (GSLs) species between controls (*n* = 30) and patients (*n* = 28) serum samples, with data separated by sex (female and male). Data are presented as mean ± REM (\**p* < 0.05, \*\**p* < 0.01, \*\*\**p* < 0.001); **b** Non-parametric, two-sided multiple Mann–Whitney *U*-test results comparing sialylated GSL subclasses between controls (*n* = 30) and patients (*n* = 28) serum samples, with data separated by sex (female and male). Data are presented as mean ± REM (\**p* < 0.05, \*\**p* < 0.01, \*\*\**p* < 0.001); **c** Non-parametric, two-sided Mann–Whitney *U*-test results comparing neutral GSL subclasses between controls (*n* = 30) and patients (*n* = 28) serum samples. Data are presented as mean ± REM (\**p* < 0.05, \*\**p* < 0.01, \*\*\**p* < 0.001); **d** Volcano plot illustrating the differential expression of neutral GSLs in Parkinson's Disease. p-values were determined using a non-parametric, two-sided Mann–Whitney *U*-test. Source data are provided as a Source Data file.

tissue-specific expression pattern: ganglio-series GSLs are predominant in the brain, lacto-series are prominent in secretory organs, neolacto-series are common in hematopoietic cells, and globo-series are abundant in erythrocytes[2]. Therefore, the deep profiling of serological GSLs of different series will allow a better footprinting of multi-organ and/or multi-cellular glycosphingolipidome in blood. Understanding these processes will significantly contribute to the discovery of biomarkers, understanding disease mechanisms, and developing treatments for diseases.

It is worth noting that we used the same LC-MS analytical conditions as for global lipidomics[32], enabling us to examine both sialylated GSLs and the overall lipid profile without requiring any modification in instrumentation.

Finally, the 4D-glycosphingolipidome and the hence established database contain entries for 376 GSL species with experimentally measured CCS values, out of which 364 also contain MS/MS spectra. This is not only in its current version a rich resource for extending the profiling capability and coverage of glycosphingolipidome in various diseases, but also as a basis to expand the annotation software resource with predictive values for ganglio- and neolacto-series that could be used also for other biospecimens, tissues and cells from humans and other organisms. In this context, we hope that the basic spectral library for GSLs and CCS values that we release here at MassIVE repository [https://doi.org/10.25345/C53J39C93], Research Tools [https://www.unimedizin-mainz.de/lipidomics-unit/lipid-research/research-tools.html], github [https://github.com/dtots17/Lipidomics-Spectral-Database.git], and Zenodo [https://doi.org/10.5281/zenodo.14882123] will aid lipidomics and MSI community to identify GSLs in prospective studies and expand analytical portfolio for GSLs.

We do anticipate that with continuous progress in technological development of robotized platforms and protocols for enrichment and fractionation, as well as in procurement of internal standards, the throughput and quantification capabilities will be expanded in the near future. The limitation of the current 4D-glycosphingolipidomics platform resides in the lack of provision of linkage types, such as α2-3 and α2-6, ß1-3 or ß1-4, etc., or discrimination of monosaccharide isomers, GalNAc versus GlcNAc, etc. These are all essential structural elements in the determination of glycan epitopes/antigens of several series of gangliosides: a, b, and c, which are particularly relevant for immune diseases and conditions. The procurement of reference standards for such structures will be pursued in the future to unequivocally determine their presence in human serum. Further adaptation of LC conditions and/or glycan-based separation of GSLs using chromatographic media such as HILIC and/or normal phase could expand the provision of CCS and MS/MS spectra for further granular dissection of the structural architecture of sialylated GSLs.

## Parkinson's disease

Serological profiling of glycolipids in general, and of sialylated GSLs in particular, in neurodegenerative and neurological diseases is rarely reported, despite the tremendously important role GSLs play in glycosynapse formation[50] and cell–cell communication in the brain. It also holds true for other diseases as well, such as immune, infectious diseases, and cancer. This is surely due to the technological difficulties emphasized here.

The capability of our 4D-glycosphingolipidomics platform for extended coverage of sialylated glycosphingolipidome and reproducible semi-quantification opened a window into the dysregulation of sialylated and global glycosphingolipidome with PD.

It is noteworthy that while studies have shown a reduction in ganglioside GM1, GD1, and GT1b levels in the substantia nigra of PD patients compared to controls[46,51], our finding revealed an increase in GM3, GM2, and GD1 in PD sera. It could be hypothesized that the blood-brain barrier disruption in PD may lead to the shedding of GSLs from the brain, resulting in their increase in the periphery. However, notably, the entire biosynthetic pathway regulated by ST3GAL5 and B4GALT1-4 (Fig. 7) is increased, suggesting also a specifically regulated pathway of chain extension from GM3 to GD1.

One of the most surprising findings is that sialylated nLc Neu5Ac-nLc6Cer and Neu5Ac-nLc4Cer species, which are predominantly found in peripheral organs, circulation, and are abundant in hematopoietic cells[2] are significantly altered with PD. It is possible that PD impacts blood cells/immune cells and/or vice versa. Altered peripheral immunity and inflammation is a hallmark of many diseases, including consequential diseases of PD. Peripheral immunity is a contributing factor to altered neuroinflammation, whereby periphery-to-brain infiltrating microglia and/or other cells maintain and/or potentiate neuroinflammation, which in turn contributes to aggravation of neurodegeneration[52,53]. On the other hand, it is here observed that ganglio-series as well as neolacto-series alterations are both underscored by a carbohydrate chain extension in their respective biosynthetic pathway. This observation is also supported by the significant increase in PD of neutral GSLs with extended glycan chains: Hex2-, Hex3-, and HexNAcHex3Cer, serving as substrates for sialylated GSL synthesis. In fact, group differences between PD and controls seem to exhibit a "systemic increase" character where 41 sialylated GSLs are dysregulated. It is tempting to inquire whether the termination of biosynthesis, as effected by sialylation of the terminal monosaccharide unit, is impaired, rendering carbohydrate chain elongation with PD. Maybe this is one of the mechanisms by which both glycosynapses, neuro- and peripheral immunity, are dysregulated with Parkinson.

Sex has a key role in PD incidence, prevalence, and mortality risk, with a male-to-female ratio of ~2:1, but the sex differences tend to disappear in advanced age. The reason for the increased vulnerability in men is unknown, but the possible protective role of female hormones has been suggested[54]. Here, interesting is that females and males have a differential dysregulation pattern of sialylated GSLs that is accentuated with PD but present in controls too. Besides proving that the platform reliably identifies clinical phenotypes, the specification of the GSL structures that discriminate the PD due to sex is valuable to infer differential pathways affected in males and females with PD, which in turn can serve as a basis to tailor the therapeutic and monitoring approaches. Since females and males experience different courses of PD progression, biomarker sets or scores tailored towards these differences should expedite the precision of risk assessment, early diagnosis, and follow-up monitoring.

Certainly, these findings require further validation in larger cohorts and translation in experimental models of PD to define the mechanistic role these species play in the etiology and development of PD, including the corresponding enzymes that regulate their synthesis and expression, and their ligands. Nevertheless, our pilot study offers novel hypotheses and starting points for further functional GSL studies and mechanistic evaluations in PD disease.

Finally, the 4D-glycosphingolipidomic platform is of general utility for disease-associated GSLs profiling at extended coverage and sensitivity of the structural heterogeneity of glycophingolipidome in biospecimens.

## Methods

### Ethics statement

For the PD study, informed consent was obtained from all human study participants, and the study was approved by the Ethical Committee of the University Medical Center of Mainz, Rheinland-Pfalz, Germany, under project number 837.311.12(8412-F).

Additionally, sera from two co-author volunteers were collected and exclusively used for the 4D-glycosphingolipidomics platform development, i.e., for the optimization of the extraction method, generation of glycosphingolipid (GSL) database library, and the proof-of-concept for the 4D-RP-LC-TIMS-PASEF analysis. The use of volunteer sera exclusively for method development complied with local ethical regulations for human sample use and adhered to the principles of the Declaration of Helsinki. Volunteer co-authors provided written informed consent for blood sample collection and agreed to the anonymized inclusion of their blood glycosphingolipidome raw data in the data repository. No compensation was offered to the co-authors.

### Nomenclature of glycosphingolipids

Nomenclature of GSLs follows the IUPAC-IUBM[55] recommendations and the shorthand nomenclature according to Liebisch et al.[56]. Briefly, GSLs are classified based on their glycan moieties, such as GM, GD, and GT, representing mono-, di-, and tri-sialylated gangliosides, respectively. Additionally, nomenclature incorporates the ceramide structure, specifying the sphingoid base (e.g., 18:1;O2) and fatty acyl chain (e.g., 18:0).

### Materials

All chemicals and solvents used in the analytical workflow were LC-MS grade. Water, methanol (MeOH), chloroform ($CHCl_3$), 2-propanol (IPA), formic acid (FA), trimethylamine, and ammonium formate were obtained from Merck (Germany). Calcium chloride ($CaCl_2$) and Trizma hydrochloride used in human serum extraction were purchased from Merck (Sigma-Aldrich, USA). Phospholipase C was acquired from Merck (Calbiochem, USA). NIST serum SRM 1951c was purchased from Sigma-Aldrich (National Institute of Standards and Technology, USA). For external calibration of the timsTOF pro mass spectrometer (Bruker Daltonics, Germany), ESI-L Low Concentration Tuning Mix (Agilent Technologies, Germany) was used.

### Glycosphingolipid standards

**Glycosphingolipid standards for 4D-glycosphingolipidomics method development and in-house library generation.** Ganglioside monosialodihexosylganglioside (GM3), monosialotetrahexosylganglioside (GM1), disialodihexosylganglioside (GD3), a-series disialotetrahexosylganglioside (GD1a), b-series disialotetrahexosylganglioside (GD1b), b-series trisialotetrahexosylganglioside (GT1b), b-series tetrasialotetrahexosylganglioside (GQ1b), C16 glucosyl (ß) ceramide (GlcCer) 18:1;O2/16:0, C16 galactosyl (ß) ceramide (GalCer) 18:1;O2/16:0, C16 lactosyl (ß) ceramide (LacCer) 18:1;O2/16:0, and Total Ganglioside Extract from Porcine Brain were obtained from Merck (Avanti Polar Lipids, Inc., USA). Ganglioside monosialomonohexosylganglioside (GM4) and disialotrihexosylganglioside (GD2) were obtained from Merck (Calbiochem, USA). Ganglioside monosialotrihexosylganglioside (GM2) was obtained from Merck (Sigma-Aldrich, USA). Each ganglioside standard was dissolved in MeOH to a 1 mg/mL stock solution and kept at −20 °C until further use. The working aliquots for each standard were prepared at a final concentration of 5 pmol/µL by diluting the stock solution with MeOH/water (8:2, v/v).

### Table 2 | Controls and Parkinson's Disease patients' information

|  | Control (n = 30) | Parkinson's Disease (n = 28) |
|---|---|---|
| Female (%) | 46.67 | 42.86 |
| Male (%) | 53.33 | 57.14 |
| Age | 62.08 ± 9 | 62.21 ± 13.21 |

Customized standard mixture consists of GM4, GM3, GM2, GM1, GD3, GD2, GD1a, GD1b, GT1b, and GQ1b at a concentration of 5 pmol/µL.

**Internal standards (ISTDs) for quantification.** C18-d5 mono-sialodiahexosylganglioside (GM3 18:1;O2/18:0-d5), C18-d5 mono-sialotetrahexosylganglioside (GM1 18:1;O2/18:0-d5), and C17 disialotetrahexosylganglioside (GD1a 18:1;O2/17:0) were obtained from Merck (Avanti Polar Lipids, Inc., USA). C16-d9 mono-sialotrihexosylganglioside (GM2 18:1;O2/16:0-d9) and C18-d3 disialodihexosylganglioside (GD3 18:1;O2/18:0-d3) were purchase from Biomol (Cayman Chemical, USA). Each internal standard was dissolved in MeOH to a 100 µg/mL stock solution and kept at −20 °C until further use.

An ISTD mixture containing 20 µg/ml GM3 18:1;O2/18:0-d5, GM1 18:1;O2/18:0-d5, GD1a 18:1;O2/17:0, GM2 18:1;O2/16:0-d9, and GD3 18:1;O2/18:0-d3, respectively, in MeOH was prepared to be spiked into samples.

### Human serum samples for method development

Sera from volunteer co-authors and NIST 1951c were used for method development. The protocol for serum collection from volunteers was described in Lerner et al.[32]. For serum preparation, blood samples were collected in 7.5 mL serum monovettes. The serum monovettes were centrifuged at 4 °C, 2000 × g for 10 min. The upper serum phase was collected and transferred into 1 mL brown Eppendorf tubes. The serum samples were stored at −80 °C until extraction.

### Clinical samples

Sera of PD patients (n = 28) along with age-matched control sera (n = 30) were provided by Movement Disorders, Imaging and Neuro-stimulation, Department of Neurology, University Medical Center Mainz, Germany (Table 2). The study participants provided informed consent, and the study was approved with project Nr. 837.311.12(8412-F) by the Ethical Committee of the University Medical Center of Mainz, Rheinland-Pfalz, Germany.

### Glycosphingolipid extraction from human serum

**Lipid extraction.** 300 µL of serum samples spiked with 12.5 µL ISTD mixture underwent overnight freeze-drying at −4 °C in glass vials using a refrigerated Centrivap Concentrator (Labconco, USA). This step ensured control over the total extraction volume. Serum lyophilization also addressed variability in serum quality and volume during sampling across cohorts, aiding in accurate recalculation of ISTDs concentration. The lyophilized sera were then extracted twice with 1.3 mL $CHCl_3$/MeOH/water (30:60:8, v/v/v). This was followed by mixing, ultrasonic treatment for 30 min, and centrifugation at 5000 × g and 4 °C for 4 min using a Sigma 4-16KS refrigerated centrifuge (Sigma Laborzentrifugen GmbH, Germany). The supernatants from each extraction step were pooled and dried under a stream of nitrogen.

**Depletion of phospholipidome.** The extracted lipids were emulsified in 500 µL Trizma hydrochloride buffer at pH 7.5, containing 10 mM $CaCl_2$, through a mild ultrasonic treatment and then were digested in 100 µL Trizma hydrochloride buffer at pH 7.5, consisting 10 mM $CaCl_2$ and 2.5 U of Phospholipase C from Bacillus cereus (Calbiochem, USA).

Afterwards, the samples were incubated in a thermoshaker at 37 °C for 16 h.

**Reversed-phase purification.** One milliliter Hypersep C18 SPE columns (50 mg, particle size 40–60 μm) (Thermo Fisher Scientific, Germany) were installed in a self-modified Nucleovac 96 Vacuum Manifold (Macherey-Nagel, Germany), which featured a plexiglass plate with 24 wells designed to accommodate both the Hypersep C18 and 3 mL glass SPE column (Macherey-Nagel, Germany). Each column was equilibrated with 6 mL of MeOH and 6 mL of water before loading the digests directly onto the column. The flow-through fraction was collected in the same vial and reapplied twice after mild ultrasonic treatment. Subsequently, the column was washed with 6 mL of water, and the polar lipids were eluted using MeOH. The collected MeOH fractions were then dried under a stream of nitrogen prior to further use.

**Silica-fractionation.** Three milliliter glass Chromabond SPE columns (Part number: 730171) (Macherey-Nagel, Germany), packed with 100 mg silica gel stationary phase (pore size 90 Å, particle size 0.063–0.200 mm) (Supelco, Germany), were attached to the self-modified vacuum manifold. Silica gel with a similar pore size was used in place of Iatrobeads, which are no longer commercially available. The columns were conditioned sequentially with 6 mL each of CHCl$_3$, CHCl$_3$/MeOH (1:1), MeOH, and CHCl$_3$. The polar lipid fraction obtained from the RP purification step was dissolved in CHCl$_3$ using mild ultrasonic treatment and applied onto the silica column. The flow-through fraction was collected and reapplied twice. After that, the column was washed with 3 mL CHCl$_3$ followed by 3 mL CHCl$_3$/MeOH (9:1, v/v). The polar fraction acquired during RP purification underwent further separation into two fractions by eluting it with 3 mL of CHCl$_3$/MeOH (2:1, v/v) for fraction 1; followed by 3 mL CHCl$_3$/MeOH (1:2, v/v), 3 mL of CHCl$_3$/MeOH (1:4, v/v), 3 mL MeOH as well as 3 mL MeOH/water (4:1, v/v) which were pooled in fraction 2. All collected fractions were dried under a stream of nitrogen and stored at −20 °C before further use.

The fractions were dissolved in 50 μL MeOH/water (4:1, v/v) for 4D-glycosphingolipidomics measurement.

### 4D LC-TIMS-PASEF MS
**Reversed-phase liquid chromatography.** GSLs were separated using a C18 Kinetex column (100 × 2.1 × 2.6 μm) (Phenomenex, Germany) on an Elute UHPLC system (Bruker Daltonics, Germany). The column compartment was maintained at 45 °C throughout the measurement. The solvent system used for separation was composed of (A): MeOH/water (1:1, v/v) and (B); MeOH/IPA (2:8, v/v), both containing 0.1% formic acid (FA), 7.5 mM ammonium formate, and 0.1% trimethylamine (TEA) in negative ion mode. For positive ion mode, the same solvent system without TEA was used. The RP-LC method involved a total separation time of 20 min with a constant flow rate of 0.2 mL/min with the following gradient: from min 0 to 1 min the mobile phase A was at 60% and mobile phase B was at 40%; the mobile phase B was then increased to 90% till min 16 and then sharp up to 99% till min 16.5; mobile phase B was decreased back to 40% till min 20. Throughout the analysis, the temperature of the autosampler was maintained at 4 °C. 20 μL of the sample extracts were injected into the LC column using partial loop injection mode.

**TIMS-PASEF.** The Elute UHPLC was coupled to timsTOF PRO and timsTOF fleX (Bruker Daltonics, Germany). The timsTOF PRO instrument was operated in negative ion mode and timsTOF flex in positive ion mode, with samples running in parallel on both instruments whenever both ion polarities were required for analysis. GSL standards were used to optimize the timsTOF PRO and timsTOF FleX parameters for ionization and detection which were set as follows: (a) ESI source

was operated in negative ion mode and positive ion mode and adjusted to following parameters: End plate offset was set to 500 V. Capillary voltage was at 3600 V in negative ion mode and 4500 V in positive ion mode. The nebulizer gas was at 2.5 Bar. Nitrogen as the dry gas was purged at the rate of 6 L/min at a temperature of 200 °C. (b) The scan mode was set to PASEF. Here, the accumulation and ramp times were each set to 100 ms. The system was operated at a 100 percent duty cycle when the accumulation and ramp times were equal. MS and MS/MS spectra were obtained in the range $m/z$ 200–2000. The mobility of ions was measured between 0.55 and 1.88 Vs/cm$^2$ in negative ion mode and between 0.55 and 1.90 Vs/cm$^2$ in positive ion mode. Within a 2 Da isolation window, precursors for data-dependent acquisition were isolated and fragmented at 50 eV collision energy in both negative and positive ion modes. Several collision energy values were tested, ranging from 30 to 90 eV, to select the best suited to generate informative fragmentation spectra for all the singly and multiply charged ions of the different GSL structures. The 0.32 s acquisition cycle included one complete TIMS-MS scan and two PASEF MS/MS scans. Low-abundance precursor ions with an intensity more than a threshold value of 100 counts but less than 4000 counts were repeatedly scheduled and excluded dynamically for 0.1 min. The TIMS was linear calibrated before the measurement using six selected ions from the Agilent ESI LC-MS tune mix [$m/z$, 1/K0: (301.9981, 0.6690 Vs/cm$^2$), (601.9790, 0.8824 Vs/cm$^2$), (1033.9881, 1.2582 Vs/cm$^2$), (1333.9689, 1.4073 Vs/cm$^2$), (1633.9498, 1.5797 Vs/cm$^2$), (1933.9306, 1.7479 Vs/cm$^2$)] in negative ion mode and [$m/z$, 1/K0: (322.0481, 0.7363 Vs/cm$^2$), (622.0290, 0.9915 Vs/cm$^2$), (922.0098, 1.1986 Vs/cm$^2$), (1221.9906, 1.3934 Vs/cm$^2$), (1521.9715, 1.5685 Vs/cm$^2$), (1821.9523, 1.7407 Vs/cm$^2$)] in positive ion mode. The online calibration in the first segment of the sequence was turned off and remained off for all sample analyses. At the beginning of each batch, however, online calibration was used with quality control (QC) samples; GSL standard mix and a lipid mixture QC, which is regularly used in-house to monitor and ensure instrument performance.

### GSLs annotation and generation of the in-house database library
The customized ganglioside standard mix, which consisted of GM4, GM3, GM2, GM1, GD3, GD2, GD1a, GD1b, GT1b, and GQ1b was measured in triplicates in both negative and positive ion modes to retrieve the 4D-descriptors for each species: $m/z$, CCS, RT, and MS/MS spectra, e.g., PASEF. Additionally, GD1a, GD1b, and gangliosides from porcine brain extract standards were analyzed in negative ion mode, and GlcCer 18:1;O2/16:0, GalCer 18:1;O2/16:0, and LacCer 18:1;O2/16:0 were analyzed in positive ion mode individually. The acquired data from a customized mixture of glycosphingolipid standards, GD1a, GD1b, gangliosides from porcine brain extract, GlcCer 18:1;O2/16:0, GalCer 18:1;O2/16:0, and LacCer 18:1;O2/16:0 standards, were subsequently uploaded to Metaboscape 2021b (Bruker Daltonics, Germany) in seven bucket tables, respectively. The processing method was as follows: (i) feature detection was performed using an intensity threshold of 100 counts, (ii) recursive feature extraction was conducted for features found in at least 1 out of 3 analysis, (iii) the feature was only included in the bucket tables if it was found in 1 analysis after recursive feature extraction. For annotation, the bucket tables were first annotated with MSDIAL-TandemMassSpectralAtlas-VS68. Next, precursor ions in negative ion mode containing fragments at nominal $m/z$ 290 in their MS/MS spectra were filtered, structurally annotated, and elucidated manually for gangliosides. In positive ion mode, precursor ions with fragments at nominal $m/z$ 274 were filtered for ganglioside identification. The PASEF spectra in positive ion mode were further used to determine the structure of the ceramide moiety. For neutral GSLs, precursor ions exhibiting a neutral loss of 162 (GlcCer and GalCer) or 342 (LacCer) were filtered and manually curated based on their PASEF spectra for structural characterization. The identified GSL species were then added to the in-house library

and further used for GSL annotation in serum and clinical samples. Average CCS and RT values of each species determined from the triplicate measurements of the above standards were added to the library. Tolerance values and scoring adjustments for automatic annotation of glycosphingolipids include: $m/z$ between 2 and 5 mDa, RT between 0.1 and 0.5 min, mSigma between 10 and 500, MS/MS score between 500 and 900, and CCS between 0.1% and 2.0%.

The library generated using the above standards is named *4D-sialoGSL standards library* and *4D-neutralGSL standards library*, and referred to as such throughout the manuscript.

## Profiling and annotation of human serological glycosphingolipidome

NIST serum SRM 1951c and serum extracts from volunteers[32] were used to investigate the glycosphingolipidome in human serum. Each serum sample was extracted and analyzed three times using the 4D-Glycosphinglipidomics workflow. The acquired data were then uploaded into Metaboscape 2021b (Bruker Daltonics, Germany) for analysis. The processing method was similar as for the analysis of GSL standards (see above Generation of in-house database library). Serum GSLs in the bucket table were annotated first using the *4D-sialoGSL standards library* and manually curated by matching their $m/z$ values to a list of theoretical $m/z$ values that encompass GSLs found in human serum[12,17] and/or referring to known biosynthesis of GSLs. Subsequent structural annotation and confirmation were done using PASEF spectra. Diagnostic fragment ions for glycosidic bond cleavage of the precursor ions and their counterparts were used to ascribe the glycan structure. Modifications such as *O*-Acetylation of sialic acid were delineated from other possible modifications in the primary structure by diagnostic ions at nominal $m/z$ 332; similarly, Neu5Gc-containing species were dissected from other possible isobars (i.e., due to ceramide composition) by the ions at nominal $m/z$ 306 specific for glycolyl-content of sialic acid.

Sulfatides, which are detectable in fraction 1 following the microextraction of GSL (see above) were annotated from PASEF spectra based on biosynthetically possible structures and inferring glycosidc bond cleavages from precursors (Fig. 4e). Here, for example the loss of sulfated group and/or sulfated monosaccharide from the precursor was used as inquiry.

Fragmentation in positive ion mode of sialylated GSLs (Fig. 1d, Supplementary Figs. 2, 5 and 6), sulfatides (Fig. 4e), and neutral GSLs (Fig. 4d) was also used to denote the ceramide composition, using ions specifically formed by sphingoid base, such as at nominal $m/z$ 262 and 264 for 18:2;O2 and 18:1;O2, respectively, and counterpart ions formed by neutral loss of fatty acyl chains of ceramide.

Their structures were also confirmed using differences in RT and CCS values for homologs series of GSLs whenever feasible. Finally, average values of the RT and CCS descriptors of identified GSLs were used to generate three in-house database libraries for human serum analysis, called *4D-sialylated GSL serum library, 4D-neutral GSL serum library, and 4D-sulfatides serum library*.

## Quantification strategies for sialylated glycosphingolipids in human serum

Six inter-day extract replicates from an individual volunteer's serum were used to test different quantification methods. Descriptor values in Metaboscape from identified GSLs were used to generate the list for Skyline 23.1.0.280 and extract ion chromatograms of GSLs for quantification purposes. Given the difficulties encountered in processing data for quantification purposes on Metaboscape 2021b, where recursively picked peaks of analytes and ISTDs, particularly for the multiply charged ions, were often outliers, we opted to quantify the identified GSLs using Skyline. The relative quantification was solely conducted utilizing the MS1 peak areas from extracted ion

chromatograms of each individual GSL species. The quantification was carried out by exporting the peak areas of the corresponding GSLs from Skyline to an Excel spreadsheet. We normalized the peak area of each analyte to the peak area of the corresponding ISTD selected as per three different sub-strategies outlined in Fig. 6b. The three sub-strategies, pertaining to different combinations of ISTD/GSL class, were used to evaluate ISTD performance in conjunction with the semi-quantitative strategies below. As the extraction method involved fractionation into two fractions, we had to account for the distribution of ISTDs between these fractions. We used the following strategies for relative quantification of GSLs:

**Strategy I—semi-quantification using average partition of internal standards between two fractions in replicate control sera.** ISTDs were added in a known quantity, allowing the calculation of ISTD amounts in each fraction. This approach involved calculating the amount of ISTDs using the average distribution percentage derived from each ISTD peak area, determined from six replicate extracts and measurements of fractions 1 and 2 in volunteer sera. The distribution of ISTDs between the two fractions was determined by considering the peak area of ISTDs in each fraction from every sample. GSLs were relatively quantified as follows:

$$A = \frac{\text{peak area of ISTD in fraction 2}}{\text{peak area of ISTD in fraction 1} + \text{peak area of ISTD in fraction 2}} \times 100 \tag{1}$$

where $A$ = percentage of ISTD in fraction 2 in each individual sample.

$$B = \text{amount of spiked ISTD(nmol)} \times \bar{A}/100 \tag{2}$$

where $B$ = amount of ISTD in fraction 2 (nmol) and $\bar{A}$ = average percentage of ISTD in fraction 2 from six measurements.

$$C = \frac{\text{Peak area of analyte}}{\text{Peak area of ISTD}} \times B \tag{3}$$

where $C$ = amount of analyte in fraction 2 (nmol).

**Strategy II—semi-quantification using the partition of internal standards between two fractions in each individual sample.** This strategy involves calculating the ISTDs using the distribution percentage from each sample. GSLs were relatively quantified as follows:

$$A = \frac{\text{peak area of ISTD in fraction 2}}{\text{peak area of ISTD in fraction 1} + \text{peak area of ISTD in fraction 2}} \times 100 \tag{4}$$

where $A$ = percentage of ISTD in fraction 2 in each individual sample.

$$B = \text{amount of spiked ISTD(nmol)} \times A/100 \tag{5}$$

where $B$ = amount of ISTD in fraction 2 in each individual sample (nmol)

$$C = \frac{\text{Peak area of analyte}}{\text{Peak area of ISTD}} \times B \tag{6}$$

where $C$ = amount of analyte in fraction 2 (nmol)

**Combination of strategies I and II.** The relative amount of GSLs was calculated using strategy I. Strategy II was used for correction when outliers were detected.

**Quantification and detection performance.** LLOD and LLOQ of sialylated GSL were determined following the guidelines for bioanalytical method validation[57]. Eight dilution points of customized standard mixture, ranging from 0.039 pmol/μL to 5 pmol/μL, were prepared, with each point spiked with 12.5 μL of ISTD mixture ISTD mixture containing 20 μg/ml GM3 18:1;O2/18:0-d5, GM1 18:1;O2/18:0-d5, GD1a 18:1;O2/17:0, GM2 18:1;O2/16:0-d9, and GD3 18:1;O2/18:0-d3. A linearity curve was generated by plotting the concentration ratio (concentration of analyte/concentration of ISTD) against the response ratio (peak area of analyte/ISTD) to determine the LLOD and LLOQ. The acceptance criterion for $R^2$ for the linearity is set to ≥0.95 due to the absence of an internal standard for each GSL subclass. However, with the exception of GM3 18:1; O2/18:0 with $R^2$ at 0.9774, all other exhibited $R^2$ values above 0.98. Subsequently, GSLs were quantified using a class- and cross-class ISTD. The chosen internal standards were: GM1 18:1;O2/18:0-d5 for GM1, GM2 18:1;O2/16:0-d9 for GM2, GM3 18:1;O2/18:0-d5 for GM3, and GD3 18:1;O2/18:0-d3 for GD3, GD2, GD1, GT1b, and GQ1b (sub-strategy 3, Fig. 6b).

LLOD and LLOQ were calculated as follows:

$$LLOD = 3.3\frac{\sigma}{s} \tag{7}$$

$$LLOQ = 10\frac{\sigma}{s} \tag{8}$$

Where $\sigma$ = standard deviation of responses and $s$ = slope of calibration curve.

The potential carry-over effect of sialylated GSLs was evaluated using the highest concentration in the dilution series (5 pmol/μL). A blank sample was analyzed after injecting the customized standard mixture to assess the carry-over of the most abundant analyte in each standard. Peak areas of analytes in the standard and blank samples were compared. Similarly, the carry-over of abundant sialylated GSL species in each subclass in serum was examined by analyzing blank samples after measuring sialylated GSLs in fraction 2 from three extract replicates. Average peak areas of sialylated GSLs in the serum extracts in fraction 2 and the blank sample were compared to determine the carry-over effect. The average values of carry-over effects as 1.2 % and were considered insignificant.

## Clinical application in Parkinson's disease

28 serum samples from individuals with PD and 30 serum samples from controls underwent analysis using the 4D-glycosphingolipidomics platform. Fraction 1 was analyzed in positive ion mode for neutral GSLs and sulfatides, while fraction 2 was analyzed in negative ion mode for sialylated GSLs. The data obtained were then uploaded to Metaboscape 2021b for automatic GSL annotation and subsequently to Skyline 23.1.0.280 for quantification purposes. Quantification of sialylated GSLs was performed using the third normalization strategy (Fig. 6b) and using the average percentage partition of ISTDs between two fractions, e.g., Strategy II (refer to "Quantification strategies for sialylated glycosphingolipids in human serum"). In cases where outlier samples were identified, the partition of ISTD between the two fractions within each individual sample was determined and employed for correction purposes (see above "Semi-quantification using the partition of internal standards between two fractions in each individual sample").

GM3 18:1;O2/18:0-d5 was used for the semi-quantification of neutral GSLs and sulfatides in fraction 1 of serum samples from PD and controls. Semi-quantitative analysis was conducted using peak area values obtained from Metaboscape 2021b software. The peak areas of neutral GSLs and sulfatides were normalized to the peak area of GM3 18:1;O2/18:0-d5.

## Statistical analysis

For the statistical analyses with the R programming language, linear models under the widely-used Limma method, which is commonly applied to evaluate differential expression between biological conditions in gene expression microarray and RNA-seq data, were used. Limma allows to evaluate the shared information across samples due to its capability to analyze entire experiments as an integrated whole rather than making piece-meal comparisons, and further employs empirical Bayesian techniques to deliver reliable differential expression predictions, which is especially advantageous when the number of sample replicates is limited, while also improving biological interpretability of the resulting differential expression of genes/features. Significantly different GSLs between groups were identified using a 2-by-2 factorial model at an adjusted $p$ value < 0.05 and corrected for multiple comparisons using the false discovery rate (FDR) method. We further adjusted the model for the effects of sex and age.

A non-parametric, two-sided Mann–Whitney $U$-test was conducted using GraphPad Prism 9.5.1 software to investigate the differences between PD and controls. For sialylated GSLs, participants were stratified by sex into male and female groups, and the test was applied separately for controls and PD within each sex group. For neutral GSLs and sulfatides, the non-parametric, two-sided Mann–Whitney $U$-test was conducted without sex-based stratification to compare between controls and PD. Statistical significance was defined for $p$ value < 0.05, with the following annotations: *$p$ < 0.05, **$p$ < 0.01, and ***$p$ < 0.001.

## Reporting summary

Further information on research design is available in the Nature Portfolio Reporting Summary linked to this article.

## Data availability

The raw data of reference samples used for method development and GSL spectral libraries: for ganglio-, neolacto-, and neutral GSLs and sulfatides generated in this study have been deposited in the MASS Spectrometry Interactive Virtual Environment (MassIVE) under the identifier MSV000097015 [https://doi.org/10.25345/C53J39C93]. Spectral libraries generated and used in this study are additionally available under the following links: Research Tools [https://www.unimedizin-mainz.de/lipidomics-unit/lipid-research/research-tools.html], GitHub [https://github.com/dtots17/Lipidomics-Spectral-Database.git], and Zenodo [https://doi.org/10.5281/zenodo.14882123]. Unless otherwise stated, all processed data supporting the results of this study can be found in the article, supplementary data, and source data files. As recommended by the International Lipidomics Society, the reporting checklist from the Lipidomics Standards Initiative[58] is included in the Supplementary Materials. Source data are provided with this paper.

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

## Acknowledgements

H.G.V. was partially funded through a scholarship by the Transmed Program of the University Medical Center of Mainz. This study was financially supported by BMBF-funded projects nr: 031L0217A and 16LK0241K to L.B. and by the curATime project nr: 03ZU1202EB to L.B. Partial financial support was obtained from DFG-funded project nr: BI 1399/2-1 and BI 1399/2-2 to L.B. We thank Dhanwin Baker for the technical support with establishing the spectral libraries and Michael Plenikowski for the graphical design.

## Author contributions

H.G.V. performed all the experiments, annotated and interpreted data, and wrote the main part of the manuscript. G.G.E. performed statistical analysis and consulted on PD etiology. L.R. acquired serum for PD. D.M., D.H., and L.R. organized sample management. S.G. advised on the manuscript, led the patients and controls clinical diagnosis, supervised sample management, and provided funding to G.G.E., D.H., and D.M. L.B. conceived, designed, and supervised the study, annotated and interpreted data, provided funding for the study, and contributed to writing.

## Funding

## Competing interests

The authors declare no competing interests.
