## [Transparent Peer Review file · Nature Communications]

Extended coverage of human serum glycosphingolipidome by 4D-RP-LC TIMS-PASEF unravels association with Parkinson's disease

Corresponding Author: Dr Laura Bindila

Version 0:

Reviewer comments:

Reviewer #1

(Remarks to the Author)

The manuscript was co-reviewed by me and two younger coworkers, and we prepared one joint review report. We appreciate the high quality of this research and manuscript preparation as well. The analysis of GSL is a quite challenging task in the field of lipidomics, and this work presents the highest number of ever reported ganglio- and neolacto-series GSLs in human serum due to the very high sensitivity of the developed method based on RP-LC/TIMS-MS and a dedicated sample preparation protocol, including the use of a larger amount of sample and phospholipid depletion followed by purification. We believe that this quality and novelty of this work is sufficient for Nature Communications. Here, we provide several suggestions and comments for the authors consideration on how to improve the manuscript, but all of them are of minor character.

1/ Quantitation

There is a certain limitation in case of using the method for quantitative measurement. Quantitative analysis of these compounds is difficult, mainly due to the lack of internal standards, but also due to fully overlapping isomers, etc., as the authors themselves discuss on page 16. The main limitation is that, during the final purification step, the authors obtained two fractions that they measured separately. In these fractions, GSLs including 5 deuterated IS can be distributed depending on their polarity. It should be noted that for each sample, the distribution of these substances in fractions may vary depending on the composition of the sample, and this fact needs to be thoroughly checked. Initially, we wanted to suggest the pooling of both fractions, but authors claim on higher noise levels in such case, so it is not the solution. We feel that the quantitation approach is rather complicated and not easy to understand all details, but we are not able to find meaningful suggestion due to know difficulties. However, the development of new bioanalytical method used for the quantitation of biological samples should be always accompanied with the validation of analytical method. Here, the validation part is rather limited, for example, there is no information on the method linearity (linear dynamic range) for selected 3 internal standards, precision, accuracy, carry-over effects using blank injections, etc. We think that at least these key parameters should be measured to make sure on the conclusions based on quantitative data. The higher level of verification would be the parallel verification by an independent laboratory, but this could be rather difficult due to the limited range of laboratories focused on the quantitative analysis of GSL.

When IS from a different lipid class is used because it is not available, then it should be termed as semi-quantitation just to fairly report in line with ILS recommendations.

2/ Fatty acyl level annotation

How have the annotation of the authors verified the fatty acyl level (type of ceramide base and N-linked fatty acyl)? Did they verify it only according to CCS and retention dependencies using measurements of selected standards? The authors should carefully check the accuracy of the annotation and go through the MS/MS data to see if there is any diagnostic ion of ceramide base or N-acyl. For example, in the representative MS/MS spectra in Figures 2 and 4, such a diagnostic ion is missing. Graphs with retention and CCS dependencies are very useful, as shown in Figure 1c, and support the correctness of the GSLs identification. These graphical dependencies should also be included in the supplementary for GSLs found in human serum. For example, GM3 d18:1/10:1 in Table 1 (Sialylated glycosphingolipid species detected by micro-RP-LC-TIMS-MS analysis in human serum), where the GSLs found in human serum are listed, GM3 d18:1/10:1 is present, which does not meet the retention dependence for GM3, so this lipid is probably misidentified.

The range of detected GM3 is impressive. We understand that the sensitivity of this method is higher in comparison to all previous works, but it is a little bit surprising that almost all fatty acyl from 10:0 to 26:0 including odd-carbon chains are detected. In our group, we cannot detect them. Another surprising thing is that the vast majority of reported lipid species are based on 18:1 base and only a very few from the whole data set are 18:0. It is well known that 18:1 is dominating, and other combinations are less common and less abundant, but the relative portion of non-18:1 seems to be extremely low. In our work, we have detected various sphingolipids based on 17:1 or 19:1 at lower concentrations in relation to 18:1, so I would expect that a similar situation could occur for GSL. It is only a comment for eventual consideration.

3/ Other GSL classes

Another question is why other GSLs, i.e., sulfatides, and neutral GSLs, are not detected. Is their sample preparation method selective only for sialylated GSLs or are other GSLs also present and the authors do not discuss them?

4/ Lipid nomenclature

We recommend the following shorthand nomenclature according to Liebisch et al. J. Lipid Res. 61 (2020) 1., similarly as Lipid MAPS and ILS. For example, GM3 d18:1/16:0 should be written as GM3 18:1;O2/16:0, GM3 t18:1/16:0 as GM3 18:1;O3/16:0, etc. It is important to use the consistent nomenclature to avoid confusion.

5/ Abbreviation of TIMS

We are surprised by the abbreviation for TIMS with lower letters "tims". We do not see any reason for this, therefore, we suggest changing it to "TIMS", similarly as for all other abbreviations to keep consistency. We have not noticed such a use in the research literature, either.

6/ Table 1

Could you comment on why some Neu5Ac-nLc4Cer species were detected as singly charged ions and the others as doubly charged?

7/ Results, line 160

We do not think that the inclusion of the normal phase system is correct here. The accurate definition of the normal phase system in chromatography is that it is based on mobile phases containing hexane or heptane as the major component of the mobile phase with an eventual very small percentage of polar additives. It is obvious that such conditions cannot be applied for the analysis of GSL because this requires a much higher polarity of the mobile phase, including a non-negligible portion of water and often ionic additives. Then, the system with such mobile phase should be correctly termed as HILIC, not normal phase, regardless of the eventual incorrect terminology in the initial manuscripts. The second option is aqueous normal phase (ANP) chromatography, which is the term coined by Andrew Alpert and used for different type of column packing. Please make sure to use the correct terminology.

8/ Switching off TIMS, line 177

This is very unexpected and is in disagreement with our experience with TIMS or any other type of ion mobility. By physical principle, the use of ion mobility devices, including TIMS, must result in a certain decrease of sensitivity (=signal intensity), typically in the range 3-10 times. In some cases, it is accompanied by the reduction of the background noise as well, which may result in a comparable signal-to-noise ratio, in some specific cases even better S/N could be reported due to increased selectivity. It would be better to phrase this sentence accurately.

9/ Database library, lines 256-265

This is a really valuable thing to create this type of library. I am sure that many people from the scientific community would be very interested to have the possibility to use it as well, therefore, my question is whether this library is publicly available to increase the impact of your efforts. I would highly encourage to make it available for other researchers.

10/ Cetrivap, line 663

Could the authors comment on what the purpose is of using Cetrivap prior to extraction instead of using a liquid sample? Is it because of lipid stability? We suggest including a brief explanation in the main text as well because this procedure is not commonly used in the field.

11/ Reporting checklist

Please consider fulfilling the Reporting checklist for this paper, which is developed by International Lipidomics Society as a tool for researchers to fairly report all experimental details, so reviewers and future readers have a complete picture about the experimental details. It will take less than 1 hour and will generate 2 pages PDF to be published as supporting information as a way to increase the transparency in the reporting of lipidomic data.

12/ Typing errors

Line 295: It should probably be Figure 3c instead of Figure 2c.

Lines 623 – 625 - Should be series instead of serie.

Line 703 – column particle size should be in micrometers, not millimeters, the Greek letter should be typed correctly.

Reviewer #2

(Remarks to the Author)

I co-reviewed this manuscript with one of the reviewers who provided the listed reports. This is part of the Nature Communications initiative to facilitate training in peer review and to provide appropriate recognition for Early Career

Researchers who co-review manuscripts.

Reviewer #3

(Remarks to the Author)

I co-reviewed this manuscript with one of the reviewers who provided the listed reports. This is part of the Nature Communications initiative to facilitate training in peer review and to provide appropriate recognition for Early Career Researchers who co-review manuscripts

Reviewer #4

(Remarks to the Author)

Mammalian sialic acid-containing glycosphingolipids (GSLs), gangliosides, are widely distributed and especially enriched in the central nervous systems. Although "glycosphingolipidome" are becoming available using the developed tools of mass spectrometry, because of their structural heterogeneity and complexity, still it is difficult to identify their detailed structure by high-throughput technology. In this manuscript, Hung G Vo et al., developed 4-dimensional (4D)-glycosphingolipidomics technology, focusing on human serum gangliosides, using the timsTOF PRO (Bruker). They successfully identified 159 gangliosides species in human serum. It is very interesting to note that they detected a unique class of ganglioside species such as GM3 (Ac) in the serum from normal and Parkinson disease (PD) patients.

The basic method itself has been published for lipidomics before (Trapped ion mobility spectrometry and PASEF enable in-depth lipidomics from minimal sample amounts | Nature Communications. 11, Article number: 331 (2020).

The results seem to be intriguing but still several issues need to be addressed.

1. Sample preparation: Preparation of ganglioside molecules are not easy because of several reasons. Some gangliosides such as GM4 and GM3 are a bit hydrophobic compared to complex type of polysialogangliosides (water-soluble nature). Authors used normal phase silica gel column to separate gangliosides into two fractions. This method is somewhat tedious and time-consuming. Did authors try to use cold acetone to remove neutral lipids and enrich GSLs (recovered in ppt fraction)? Or, how about ion-exchange column?
2. The patient samples in Table 2. Do the PD patients carry any mutations, especially in GBA1 and PLA2G6?
3. The reviewer supposes that GM3 containing shorter acyl chains, C10:0 ~ C15:1 is very rare in human tissues. Are these endogenously formed by CERS (ceramide synthase) enzymes or from diet, or metabolites by microbiota. Is this type of GM3 isolated and purified and its structure confirmed biochemically or published before? It is difficult to detect these GSLs in human samples (Aoki et al., Clinical Mass Spectrometry 14 (2019) 190-114). As describe in the text (Figure 6), the endogenous synthesis of gangliosides is carried out by various glycosyltransferases in the Golgi apparatus. However, ganglioside metabolism, unlike that of cholesterol synthesis, has a well-developed recycling system. That is, glycosidases are also involved in the cellular composition of GSLs. In particular, GBA1 is a well-known high risk factor for PD disease. In relation to this, it would be nice if the authors analyze the levels of glucosylceramide and its molecular species (fatty acyl chains) by the timsTOF PRO.

Small concerns:

1. Discussion Page 25, line 583: What is "Glycosynapses"? The use of this term is accepted in neuroscience community?
2. There exist more polar gangliosides such as GQ1b/GQ1b-alpha gangliosides. The current MS analytical systems could determine such high molecular weight GSLs? What is the limitation of the method?
3. The method could determine NeuGc-species in the polar GSLs in the CNS. Such NeuGc-containing GSLs are detectable in human serum samples?

Reviewer #5

(Remarks to the Author)

This manuscript describes an optimized process for analysis of glycosphingolipids from serum with improvements in both extraction procedure and liquid chromatography-ion mobility-mass spectrometry analysis. The method's improvements can have high value to the community. While the test case clearly illustrates the capability of the method; the differences in GSLs are quite modest, and it is unclear that they can be used to describe differences between controls and patients with Parkinson's disease. The writing is clear, but some revisions are needed to improve the quality of the manuscript.

1. Is there a known disease that would create a significant difference in glycosphingolipids (GSL) in circulation? The comparison of Parkinson's disease to controls shows that the levels of GSL species can be measured in both, but it does not have a positive control that demonstrates that the assay recognizes expected differences in biology.
2. The order of the experiments in the description should follow the order of operations (LC first, then ion mobility, mass spectrometry, and tandem mass spectrometry). Organize the results in the order of the experiment also; it is recommended to move the results for the improvement in extraction forward before the cataloging of the GSL species in the biological samples.
3. Minor editing is needed to correct grammar.

Specific Comments

Line 84 and line 102 Ion mobility should precede mass spectrometry.

Lines 104-5 Change text from IMS to IMS-MS

Lines 107-109 Do the referenced methods for 3D-lipidomics incorporate liquid chromatography? The use of the 3D and 4D,

which is mainly a Bruker trademark, can be a little confusing for readers new to the area.

Line 119 Normal phase LC should precede MS.

Line 199 Change text to *de novo* in italics.

Figure 1A: Improve labels on the axes.

Figure 1D: Add a Key for the symbols. Are fragment ions observed in these spectra?

Figure 2: Add a key for the symbols.

Line 270 and throughout the manuscript: Change text to nanoflow replacing nanoLflow.

Line 295 The reference to Figure 2C should be 3C.

Lines 323-4 Why does microflow perform better? Is it using higher loading?

Figure 4A: Add more description in the caption for the MS data panels.

Lines 380-387 The text describing the quantification strategies is confusing; addition of a table or workflow diagram to Figure 5 would be helpful to the reader.

Figure 5: Change the y axis from % to Measured GSLs (%) for clarity. Why was 35% CV chosen? 20% CV is a more typical cutoff. A plot of the CV values for all of the GSLs would also be useful.

Figure 6b: The heat map does not show that some GSLs are systematically different between controls and Parkinson's disease patients; it shows that a couple of individual patients' samples have high expression compared to everyone else.

Figure 7: The GSL data show overlapping distributions in most cases. Despite strong p values, it does not seem like there is much separation between groups. Addition of the mean or median and standard deviation in each group will help clarify the levels of difference for the reader. Addition of Hellinger or Bhattacharya distance would also improve the ability to show whether the two groups separate from each other.

Lines 676 and 686: Do the authors mean SPE cartridges? Please provide the part numbers to support replication of results in other labs.

Line 703: Define the column dimensions (length x ID) and check particle size microns not mm.

Lines 764-5 and 780: Are the authors making the libraries of GSLs available to the public? This resource could have very strong value for the community.

Table 1 is quite long and may need to be summarized with the full version included in the supplement.

Supplemental tables should have the caption and headings on each page in the document, or just submitted as spreadsheets.

Version 1:

Reviewer comments:

Reviewer #1

(Remarks to the Author)

We have thoroughly reviewed the authors' responses and appreciate their excellent work during the revision process. Their efforts have not only addressed the feedback effectively but also further enhanced the quality of manuscript. We have no additional comments and firmly believe that this work represents a valuable contribution to the field of glycosphingolipid analysis.

Reviewer #2

(Remarks to the Author)

Reviewer #3

(Remarks to the Author)

Reviewer #4

(Remarks to the Author)

The revisited paper is carefully addressed to each inquiry and question. I think this paper will be very useful in the GSL research community.

Reviewer #5

(Remarks to the Author)

The authors have addressed the questions raised in the first review and significantly improved their manuscript. The text and methods are clear now and should aid in other investigators applying the same method. The different mechanisms of data sharing are also strong. All concerns have been appropriately addressed with thorough and well thought out answers, and the corresponding changes have been made in the manuscript.

Only one minor recommendation, add a key for the sugar residues in the figure or caption for Supplemental Figures 2, 5, and 6.

Answer to REVIEWER COMMENTS

Reviewer #1 (Remarks to the Author):

The manuscript was co-reviewed by me and two younger coworkers, and we prepared one joint review report. We appreciate the high quality of this research and manuscript preparation as well. The analysis of GSL is a quite challenging task in the field of lipidomics, and this work presents the highest number of ever reported ganglio- and neolacto-series GSLs in human serum due to the very high sensitivity of the developed method based on RP-LC-TIMS-MS and a dedicated sample preparation protocol, including the use of a larger amount of sample and phospholipid depletion followed by purification. We believe that this quality and novelty of this work is sufficient for Nature Communications. Here, we provide several suggestions and comments for the authors consideration on how to improve the manuscript, but all of them are of minor character.

Answer:

We thank the reviewer and his team for the extensive revision, feedback, suggestions, and also for the positive appreciation. We have addressed all the issues raised and suggestions, including provision of the neutral glycosphingolipids and sulfatides detected in Fraction 1, validation data for the semi-quantification, all the fragmentation spectra to support structural assignment of new compounds found, and we will make available a spectral library if and upon acceptance of manuscript.

The detailed responses to each point are outlined below.

1/Quantitation

There is a certain limitation in case of using the method for quantitative measurement. Quantitative analysis of these compounds is difficult, mainly due to the lack of internal standards, but also due to fully overlapping isomers, etc., as the authors themselves discuss on page 16. The main limitation is that, during the final purification step, the authors obtained two fractions that they measured separately. In these fractions, GSLs including 5 deuterated IS can be distributed depending on their polarity. It should be noted that for each sample, the distribution of these substances in fractions may vary depending on the composition of the sample, and this fact needs to be thoroughly

checked. Initially, we wanted to suggest the pooling of both fractions, but authors claim on higher noise levels in such case, so it is not the solution. We feel that the quantitation approach is rather complicated and not easy to understand all details, but we are not able to find meaningful suggestion due to know difficulties. However, the development of new bioanalytical method used for the quantitation of biological samples should be always accompanied with the validation of analytical method. Here, the validation part is rather limited, for example, there is no information on the method linearity (linear dynamic range) for selected 3 internal standards, precision, accuracy, carry-over effects using blank injections, etc. We think that at least these key parameters should be measured to make sure on the conclusions based on quantitative data. The higher level of verification would be the parallel verification by an independent laboratory, but this could be rather difficult due to the limited range of laboratories focused on the quantitative analysis of GSL.

When IS from a different lipid class is used because it is not available, then it should be termed as semi-quantitation just to fairly report in line with ILS recommendations.

Answer:

Indeed, the quantitation is particularly challenging when fractionation is necessary, as it is in this case. Without fractionation, the sialylated GSLs with long chain carbohydrates would not be enriched enough to detect as broad a plethora as we see. Hence, we use the partition percentage of internal standards between the two fractions determined in replicate reference serum (which is a pooled serum of volunteers, see Strategy I) to calculate the concentration of individual molecules (Fig. 6a). We also used different combination of internal standards-GSL subclass for cross-evaluation of quantitation reproducibility and we consider this as an extra internal control (Fig. 6b). We are definitely recommending the use of pooled sample (see Strategy I) for prospective studies of different biological matrices and also of different serum sources, i.e. different diseases; as indeed the matrix plays a big role in reproducibility. In fact, that is why we used the partition coefficient in replicate reference sera (Fig. 3).

Indeed, pooling of the two fractions did hinder the extensive coverage of the glycosphingolipidome. We are sure that future refinements of the method with inclusion of ever evolving enrichment/purification columns/materials, or future use of 2D-LC systems might help in analysis of a pooled fraction 1 and 2, and we hope that our work will open new venues for the community.

We rewrote the section of quantification method to better explain how the partition of internal standards between fractions was used to calculate GSL concentrations.

We made sure that we state everywhere semi-quantification. We provided the ILS reporting checklist in Supplementary Materials.

Indeed, ideally independent verification of GSL levels in reference sera and/or in disease cohorts is preferred, but as the reviewers note it is not easy to find this expertise. Although in late 90's and early 2000's GSL extraction and MS-analysis has been advanced by Peter-Katalinic's, Costello's, Müthing's labs of which some of the authors benefited from; the tedious manual spectra annotation and laborious procedure have hindered adoption of this field in the mainstream research which resulted in less current expertise in this field. We hope, however, that this work will invite new developments by the community, and we are open to participate in future initiatives of lipidomics community to include the GSLs in the harmonisation protocols. Also, we are open to host cross-laboratory GSL data validation.

2/ Fatty acyl level annotation

How have the annotation of the authors verified the fatty acyl level (type of ceramide base and N-linked fatty acyl)? Did they verify it only according to CCS and retention dependencies using measurements of selected standards? The authors should carefully check the accuracy of the annotation and go through the MS/MS data to see if there is any diagnostic ion of ceramide base or *N*-acyl. For example, in the representative MS/MS spectra in Figures 2 and 4, such a diagnostic ion is missing.

Answer: Indeed, this is an important question which we have to clarify better in the manuscript and provided additional structural data to support this.

In negative ion mode, fragment ions arising from the ceramide moiety fragmentation to sphingoid and fatty acyls are not or poorly detectable due to poor or no charge retention- this is also instrument independent in our knowledge. This fragmentation behavior is exemplified in Figure 1d, where we included the fragmentation of GD1a and GD1b in positive ion mode, in Supplementary Figure 2, 5 and 6, where PASEF spectra of GD3 39:1;O2 and GM3 18:1;O2/12:0 in positive and negative ion species

illustrate the differential fragmentation ions obtained and the diagnostic fragment ions generated from ceramide moiety in positive ion mode used to determine the ceramide content.

We used the positive ion mode analysis to elucidate the ceramide content of the sialylated GSL (Fig. 1d, Supplementary Fig. 5 and 6, Supplementary Data 1), by alignment of fragmentation spectra obtained for a give species detectable in both modes at a given RT.

Fragmentation of the ceramide moiety in positive ion mode at collision energy of 50 eV generates fragment ions corresponding to both fatty acyls and sphingoid moiety. We carried out this analysis in positive ion mode in reference mixtures, where primarily GM, GD, and a few GT species were detected as positive ions, and from their fragmentation the determination of sphingoid bases and corresponding counterparts was achieved. The data are included also in Supplementary Data 1. For GM species it is visible that sphingoid bases with d18:1 predominates the content; at lower abundance, and coeluting with isomers/isobars containing 18:1;O2 are also species with 16:1;O2, 17:1;O2, 19:1;O2. We included in Figure 1d, Supplementary Figure 2, 5, and 6 the fragmentation of positive ions of GD1, GD3 and GM3 species, respectively, for reference of structural annotation. From positive ion mode analysis of serum fraction where GM3 species were detected we determine the ceramide composition: sphingoid and fatty acyl. Based on retention time alignment between GM in positive and negative ion mode, we can annotate also the co-eluting isomers stemming from sphingoid/fatly acyl composition (Supplementary Data 4). We infer that such heterogeneity extends to longer-chain GSLs, but we kept their annotation with d18:1/fatty acyl as the dominant one. We extended this annotation for neutral glycolipids and sulfatides (see below) which are now included in the manuscript (Supplementary Data 5 and 6).

Sialylated GSL longer than GM3 in human serum, which are not detectable in positive ion mode, we annotated the 18:1;O2 sphingoid base-containing as the dominant isomer (Figure 5, Supplementary Data 2, 3, and 4) based on i) inference from GM data in positive ion mode (Supplementary Figure 5 and 6); ii) also on data newly acquired on neutral GSLs (Figure 4d, Figure 5, Supplementary Data 5), and iii) prior literature¹.

Graphs with retention and CCS dependencies are very useful, as shown in Figure 1c, and support the correctness of the GSLs identification. These graphical dependencies should also be included in the supplementary for GSLs found in human serum. For example, GM3 d18:1/10:1 in Table 1 (Sialylated glycosphingolipid species detected by micro-RP-LC-TIMS-MS analysis in human serum), where the GSLs found in human serum are listed, GM3 d18:1/10:1 is present, which does not meet the retention dependence for GM3, so this lipid is probably misidentified.

Answer: We included 3 dimensional data displays for standards (Figure 1b and c, Supplementary Figure 3 and 4), for sialylated GSL in human serum and new classes included in the manuscript: neutrals and sulfatides (Figure 5, Supplementary Figure 7). The retention time of GM3 d18:1/10:1 has been wrongly shifted in the manuscript during to copy-paste. This is now corrected (Supplementary table 4).

The range of detected GM3 is impressive. We understand that the sensitivity of this method is higher in comparison to all previous works, but it is a little bit surprising that almost all fatty acyl from 10:0 to 26:0 including odd-carbon chains are detected. In our group, we cannot detect them.

Answer: We attribute the detected range of heterogeneity due to ceramide content to the reversed-phase separation, which increases the separation efficiency of such species, reducing thus the overlapping level and ionisation suppression of lower abundant species when less chromatographically resolved (Supplementary Data 4). We included in the manuscript in Supplementary Figure 5 and 6 the fragmentation spectra of GM3 with odd-chain fatty acyls and shorter chain fatty acids: GM3 18:1;O2/12:0 and GM3 18:1;O2/14:0 containing diagnostic fragment ions for the fatty acyl and sphingoid base in these species.

Another surprising thing is that the vast majority of reported lipid species are based on 18:1 base and only a very few from the whole data set are 18:0. It is well known that 18:1 is dominating, and other combinations are less common and less abundant, but the relative portion of non-18:1 seems to be extremely low. In our work, we have

detected various sphingolipids based on 17:1 or 19:1 at lower concentrations in relation to 18:1, so I would expect that a similar situation could occur for GSL. It is only a comment for eventual consideration.

Answer:

Please refer to the answer above for the data supporting the determination of sphingoid/fatty acyl content.

In addition, indeed, in our lipidomics data of various human sources we also observe that sphingomyelins and sphingosines have a wider range of odd chain bases. But we didn't observe this for glycolipids in human samples now or in our^{2,3} and other's previous work^{1,4}.

3/ Other GSL classes

Another question is why other GSLs, i.e., sulfatides, and neutral GSLs, are not detected. Is their sample preparation method selective only for sialylated GSLs or are other GSLs also present and the authors do not discuss them?

Answer: We have originally focused our manuscript on sialylated long chain GSLs since this were/are notoriously challenging to analyze. As we have mentioned in the manuscript, fraction 1 was known to us to contain neutral GSLs³ and we now analyzed this in Parkinson's Disease and control samples as well, structurally annotated and evaluated their relative content in these two groups.

We included thus the fully range of neutral GSLs we detected and structurally annotated see in Figure 4d, Figure 5 and in Supplementary Data 5. Additionally, we detected and structurally annotated sulfatides in human serum. Fragmentation spectra representative for sulfatides are included in the Figure 4e, Figure 5, Figure 8, Supplementary Figure 5, and Supplementary Data 6. In Figure 5, the 3D cubic representation of the 72 sulfatides detected and of 145 neutral GSLs in human serum are now included. We also provided full list of CCS and RT parameters for these structures (Supplementary Data 5 and 6, Supplementary Fig. 7), and referred our coverage of neutral species to the work of Xia et coll⁵ recently published in Analytical

Chemistry Journal on detection of neutral GSLs with derivatisation, enrichment and MS approach.

4/ Lipid nomenclature

We recommend the following shorthand nomenclature according to Liebisch et al. J. Lipid Res. 61 (2020) 1., similarly as Lipid MAPS and ILS. For example, GM3 d18:1/16:0 should be written as GM3 18:1;O2/16:0, GM3 t18:1/16:0 as GM3 18:1;O3/16:0, etc. It is important to use the consistent nomenclature to avoid confusion.

Answer: Thank you for suggestion. We adapted the nomenclature accordingly.

5/ Abbreviation of TIMS

We are surprised by the abbreviation for TIMS with lower letters “tims”. We do not see any reason for this, therefore, we suggest changing it to “TIMS”, similarly as for all other abbreviations to keep consistency. We have not noticed such a use in the research literature, either.

Answer: We changed it to TIMS. In another paper of ours the reviewers asked us to keep it with lower letters. We agree it should be reverted back to capital letters.

6/ Table 1

Could you comment on why some Neu5Ac-nLc4Cer species were detected as singly charged ions and the others as doubly charged?

Answer: Both ion forms are possible for these species in general. Generation and fragmentation of multiply charged ions is preferred due to a richer fragmentation pattern for glycan structure elucidation. However, for these low abundant Neu5Ac-nLc4Cer species, the m/z range where doubly charged ions occur is much more crowded and can lead to higher suppression of very low abundant doubly charged ions and/or to missed selection of precursor for fragmentation. For these low abundant Neu5-nLc4Cer species the m/z area of singly charged ions was less populated and these ions were selected for fragmentation, while the doubly charged not. For longer chain GSL species, anyway the ion mobility values fall out of the range we set, and we

reported in the Table only the species for which we measured all 4D parameters. At the request of the Reviewer 5 we changed the Table to a smaller one with more aggregated information on GSL content of human serum, and provided all the 4D parameters in Supplementary Data 1-6.

7/ Results, line 160

We do not think that the inclusion of the normal phase system is correct here. The accurate definition of the normal phase system in chromatography is that it is based on mobile phases containing hexane or heptane as the major component of the mobile phase with an eventual very small percentage of polar additives. It is obvious that such conditions cannot be applied for the analysis of GSL because this requires a much higher polarity of the mobile phase, including a non-negligible portion of water and often ionic additives. Then, the system with such mobile phase should be correctly termed as HILIC, not normal phase, regardless of the eventual incorrect terminology in the initial manuscripts. The second option is aqueous normal phase (ANP) chromatography, which is the term coined by Andrew Alpert and used for different type of column packing. Please make sure to use the correct terminology.

Answer: We changed the reference to aqueous normal phase chromatography.

8/ Switching off TIMS, line 177

This is very unexpected and is in disagreement with our experience with TIMS or any other type of ion mobility. By physical principle, the use of ion mobility devices, including TIMS, must result in a certain decrease of sensitivity (=signal intensity), typically in the range 3-10 times. In some cases, it is accompanied by the reduction of the background noise as well, which may result in a comparable signal-to-noise ratio, in some specific cases even better S/N could be reported due to increased selectivity. It would be better to phrase this sentence accurately.

Answer: We understand the surprising effect of turning of the TIMS module. We provided electron ion chromatogram and base peak chromatogram in Supplementary Figure 1 we acquired with TIMS on and with TIMS off for the GM1 standards clearly demonstrating superior sensitivity with TIMS on. The best explanation we have for this

is what the instrument provider is claiming that through TIMS accumulation a higher ion usage and increased sensitivity can be obtained. I think evidence for superior sensitivity of TIMS technology was presented for proteomics application. We are however not able to determine from the current experiments the physical principle and parameters conducive to that.

9/ Database library, lines 256-265

This is a really valuable thing to create this type of library. I am sure that many people from the scientific community would be very interested to have the possibility to use it as well, therefore, my question is whether this library is publicly available to increase the impact of your efforts. I would highly encourage to make it available for other researchers.

Answer: Thank you for appreciating this and suggestion to open the library to the public. We have decided to include access to our standard GSL library we created that contains sialylated, neutral glycosphingolipids and sulfatides. This library we use to annotate and curate the biological samples. A link to download this library is provided in the manuscript and Material and Methods. This library will be made available at this link <https://www.unimedizin-mainz.de/lipidomics-unit/lipid-research/research-tools.html> and <https://github.com/dtots17/Lipidomics-Spectral-Database.git>, if and upon manuscript acceptance. It is worth noting, however, that the conversion of the spectral library data from Metaboscope in readable formats for other instruments/software is not supported by the provider: so we developed a script to convert the library in readable formats. We will ensure continuous support to the community to disseminate this library and expand it.

10/ Cetrivap, line 663

Could the authors comment on what the purpose is of using Cetrivap prior to extraction instead of using a liquid sample? Is it because of lipid stability? We suggest including a brief explanation in the main text as well because this procedure is not commonly used in the field.

Answer: Thank you. We added the reasoning for this. Essentially, due to the fact that GSLs are of very low abundance in the serum, prevention of extra loss due to higher total volumes (starting serum volume + extraction solvent = 1.6 mL) is beneficial. Moreover, sera volumes are not always accurately sampled across a cohort (foam formation, deposition or precipitates) and hence control of the volumes and ensuring an accurate recalculation of ISTDs concentration amount is more readily performed when sera is evaporated. Additionally, in early experience we observed less matrix effects, since resolubilisation of other biomolecules from serum lysates was less effective.

11/ Reporting checklist

Please consider fulfilling the Reporting checklist for this paper, which is developed by International Lipidomics Society as a tool for researchers to fairly report all experimental details, so reviewers and future readers have a complete picture about the experimental details. It will take less than 1 hour and will generate 2 pages PDF to be published as supporting information as a way to increase the transparency in the reporting of lipidomic data.

Answer:

We included the Reporting checklist of ILS in Supplementary Materials.

12/ Typing errors

Line 295: It should probably be Figure 3c instead of Figure 2c. **Done**

Lines 623 – 625 - Should be series instead of serie. **Done**

Line 703 – column particle size should be in micrometers, not millimeters, the Greek letter should be typed correctly. **Done**

Answer:

We adjusted accordingly and did a thorough proofing of the text.

Reviewer #2 (Remarks to the Author):

Answer: Thank you for your constructive input.

Reviewer #3 (Remarks to the Author):

I co-reviewed this manuscript with one of the reviewers who provided the listed reports. This is part of the Nature Communications initiative to facilitate training in peer review and to provide appropriate recognition for Early Career Researchers who co-review manuscripts

Answer: Thank you for your constructive input.

Reviewer #4 (Remarks to the Author):

Mammalian sialic acid-containing glycosphingolipids (GSLs), gangliosides, are widely distributed and especially enriched in the central nervous systems. Although “glycosphingolipidome” are becoming available using the developed tools of mass spectrometry, because of their structural heterogeneity and complexity, still it is difficult to identify their detailed structure by high-throughput technology. In this manuscript, Hung G Vo et al., developed 4-dimensional (4D)-glycosphingolipidomics technology, focusing on human serum gangliosides, using the timsTOF PRO (Bruker). They successfully identified 159 gangliosides species in human serum. It is very interesting to note that they detected a unique class of ganglioside species such as GM3 (Ac) in the serum from normal and Parkinson disease (PD) patients. The basic method itself has been published for lipidomics before (Trapped ion mobility

spectrometry and PASEF enable in-depth lipidomics from minimal sample amounts | Nature Communications. 11, Article number: 331 (2020). The results seem to be intriguing but still several issues need to be addressed.

Answer: Thank you for the comprehensive evaluation of the manuscript, and pertinent questions and suggestions.

Indeed, the first paper showing that TIMS-PASEF is suitable for lipidomics was published in: Nature Communications. 11, Article number: 331 (2020). Trapped ion mobility spectrometry and PASEF enable in-depth lipidomics from minimal sample amounts.

For the analysis of the GSLs we used the microflow RP-UPLC-TIMS-PASEF conditions optimized for high-throughput lipidomics in Lerner et al. Nat Commun 2023⁶.

1. Sample preparation: Preparation of ganglioside molecules are not easy because of several reasons. Some gangliosides such as GM4 and GM3 are a bit hydrophobic compared to complex type of polysialogangliosides (water-soluble nature). Authors used normal phase silica gel column to separate gangliosides into two fractions. This method is somewhat tedious and time-consuming. Did authors try to use cold acetone to remove neutral lipids and enrich GSLs (recovered in ppt fraction)? Or, how about ion-exchange column?

Answer: It is indeed true that the extraction method is a bit tedious: However, it is much less laborious and faster than the current published methods for GSL, especially for sialylated GSL from any samples. The silica-fractionation ensures enrichment of the long chain sialylated GSLs, which we were not able to see without such enrichment. We based the current optimisation of extraction on prior expertise and work in J. Glycomics and Lipidomics³.

The goal was to preserve the fraction of neutral glycolipids too and not to lose them during precipitation or other enrichment steps. From the literature that we surveyed so far and our own experience with lipidomics of different sources and with different extraction strategies, GM3 and HexCer, and Hex2Cer were the predominant species

found so far in human serum with or without precipitation. We didn't try now ion exchange chromatography for the present purpose, since we had no evidence from literature and in prior work that it ensures both enrichment of long chain sialylated GSLs and preservation of the neutral GSL fraction^{2,7}.

In fact, we now included in the manuscript the analysis of fraction 1 in PD and control samples, and of reference serum samples; where we show an extensive coverage both of neutral glycolipids and of sulfatides, thus, attesting to the utility of this fractionation (Figure 4d and e, Figure 5, Figure 8, Supplementary Figure 7, Supplementary Data 5, 6, 9, and 11).

No doubt, with the advent of affinity chromatographic materials, columns and robotic, the extraction procedure presented here can be further refined and parallelized and we hope our basic work here is a solid ground for such developments by the community.

2. The patient samples in Table 2. Do the PD patients carry any mutations, especially in GBA1 and PLA2G6?

Answer: This is an important question. The patients were not tested for these mutation since they were not clinical indication for this. However, it is known to us that such mutations are risk factors for the development of PD and that increased GBA1 activity has been determined in PD patients; although the corresponding HexCer product was not significantly altered in sera.

We, however, included here the analysis of fraction 1 containing the neutral GSLs (Figure 4d, 5, 8 and Supplementary Data 5, 9, and 11), including HexCer and downstream biosynthetic chain extension to Hex2-Hex3, HexNAcHex3-, HexNAcHex4- and FucHexNAcHex3Cer, and evaluated their levels in PD versus Control. In Figure 8, the statistical results of these classes in PD versus control are presented, showing that HexCer class or individual HexCer structures are not significantly upregulated in PD; however, downstream classes extended by glycosyltransferases to Hex2-, Hex3- and HexNAcHex3- Cer are indeed significantly increased. Since these species are substrates for sialyltransferases to generate

sialylated GSLs which we saw to be significantly altered, the findings concur well with the rest of the data. It will definitely be interesting to see, whether the lack of change in the HexCer in serum of PD with increased GBA1 activity is due to downstream synthesis of extended GSL species. In Fraction 1, we also detected sulfatides, which we now included the data for in the manuscript (Figure 4e, Figure 5, Supplementary Figure 7, Supplementary Data 6) and evaluated them in PD and control samples (Figure 8c, 8d, Supplementary Data 9 and 11) too.

3. The reviewer supposes that GM3 containing shorter acyl chains, C10:0 ~ C15:1 is very rare in human tissues. Are these endogenously formed by CERS (ceramide synthase) enzymes or from diet, or metabolites by microbiota. Is this type of GM3 isolated and purified and its structure confirmed biochemically or published before? It is difficult to detect these GSLs in human samples (Aoki et al., Clinical Mass Spectrometry 14 (2019) 190-114). As describe in the text (Figure 6), the endogenous synthesis of gangliosides is carried out by various glycosyltransferases in the Golgi apparatus. However, ganglioside metabolism, unlike that of cholesterol synthesis, has a well-developed recycling system. That is, glycosidases are also involved in the cellular composition of GSLs. In particular, GBA1 is a well-known high risk factor for PD disease. In relation to this, it would be nice if the authors analyze the levels of glucosylceramide and its molecular species (fatty acyl chains) by the timsTOF PRO.

Answer: The reviewer is correct that GM3 with short and odd chains fatty acyls are rather rare in human tissues. In fact their presence was more observed in human milk, microbiota, and their origin as dietary sources has been also evidenced⁸⁻¹⁰. Most of the structural determinations and findings of these structures in the above samples were aided or primarily done by MS. We provide here unambiguous structural proof of the fatty acyl/sphingoid content for these species (Supplementary fig. 5 and 6). Indeed, in general deep coverage of GSLs is notoriously difficult, but given the extraction and enrichment protocol combined with reversed-phase, *m/z* and ion mobility separation, we could detected and characterize these species. We are sure, other colleagues will

be able to apply and/or build-up on these experimental strategies to improve their coverage of GSL in their respective samples.

Concerning the level of HexCer: please refer to the answer above.

We included in Results, Material and Methods, and Discussion the corresponding data and discussion of findings on neutral GSLs for PD.

Small concerns:

1. Discussion Page 25, line 583: What is “Glycosynapses”? The use of this term is accepted in neuroscience community?

Answer: I am not sure if it is a term adopted by all the neuroscience community, but the term was used by Hakomori¹¹ to describe the cell-cell communication and signaling function of gangliosides which are also organized in lipid rafts-like regions in the cell membrane.

2. There exist more polar gangliosides such as GQ1b/GQ1b-alpha gangliosides. The current MS analytical systems could determine such high molecular weight GSLs? What is the limitation of the method?

Answer: Indeed, we have detected and structurally characterize GQ species in serum: Figure 4b and Figure 5; GQ1 in porcine brain extract in Figure 2d, Supplementary Table 1, Supplementary Figure 4, Supplementary data 2-4.

3. The method could determine NeuGc-species in the polar GSLs in the CNS. Such NeuGc-containing GSLs are detectable in human serum samples?

Answer: Neu5Gc- containing species are biosynthesized by animals such as mice, rats, porcine, but not by humans. Hence, we found them in porcine brain. In human samples, Neu5Gc moiety can be uptaken from dietary sources such as red meat¹² or

and can get accumulated in tissues due to an affected metabolism with disease (such as shown in cancer¹³).

Reviewer #5 (Remarks to the Author):

This manuscript describes an optimized process for analysis of glycosphingolipids from serum with improvements in both extraction procedure and liquid chromatography-ion mobility-mass spectrometry analysis. The method's improvements can have high value to the community. While the test case clearly illustrates the capability of the method; the differences in GSLs are quite modest, and it is unclear that they can be used to describe differences between controls and patients with Parkinson's disease. The writing is clear, but some revisions are needed to improve the quality of the manuscript.

Answer: We thank the reviewer for the appreciation and critical revisions suggested.

1. Is there a known disease that would create a significant difference in glycosphingolipids (GSL) in circulation? The comparison of Parkinson's disease to controls shows that the levels of GSL species can be measured in both, but it does not have a positive control that demonstrates that the assay recognizes expected differences in biology.

Answer: We appreciate this comment. It would be always ideal to have a reference sample cohort with expected GSL differences to validate assays. Procurement of samples from diseases with well established GSL pathology: Tay-Sachs, lysosomal storage diseases, Gaucher diseases-is not readily feasible for method development and validation.

Changes of GSLs level with PD are, however, expected and not surprising given the prior literature on GM3 and GBA1 pathology. In that sense, the findings on GM3 increase with PD in serum aligns with prior knowledge. We think that the lack of methods to deeply cover the glycosphingolipidome in serum and plasma at a

throughput higher than days and weeks per sample preparation is simply the limiting factor in determining the role of longer chain GSLs in disease. We hope our work will aid in such further endeavours. Also, we find it very interesting that in fact, also expected clinical differences between male and female are reflected by GSL profile of PDs as well.

We added in the manuscript, data on validation, LLOQ, LLOD, linearity, and carry-over effect (Figure 6d, Supplementary Table 3 and 4) in addition to already included replicate extraction, analysis, cross-validation of quantitation with multiple ISTD-strategies. We think that, collectively, this provides a solid evidence on the method's validity. Besides, the precursor of the extraction method and MS analysis³ also rendered robust results on cancer samples.

2. The order of the experiments in the description should follow the order of operations (LC first, then ion mobility, mass spectrometry, and tandem mass spectrometry). Organize the results in the order of the experiment also; it is recommended to move the results for the improvement in extraction forward before the cataloging of the GSL species in the biological samples.

Answer: We thank you for this suggestion. We re-ordered the description of experiments in the material and methods. We kept, however, in the results part the TIMS profiling of GSL standards before extraction, since the prerequisite for detecting extracted GSL was to ensure that with reverse-phase chromatography at microflow, and/or with the solvent system we actually had for more classical lipidomics we are actually able to detect and structurally characterize GSLs. Reversed phase chromatography is not and was not the go-to option for GSL since their high hydrophilic nature made them more suitable for aqueous normal phase and/or HILIC chromatography. So, that in and of itself was the prerequisite development.

3. Minor editing is needed to correct grammar.

Answer: We thank the reviewer for the comprehensive reading and editing suggestions. We addressed all the changes and answered the additional questions below.

Specific Comments

Line 84 and line 102 Ion mobility should precede mass spectrometry. **Done**
Lines 104-5 Change text from IMS to IMS-MS. **Done**

Lines 107-109 Do the referenced methods for 3D-lipidomics incorporate liquid chromatography? The use of the 3D and 4D, which is mainly a Bruker trademark, can be a little confusing for readers new to the area.

Answer: We exchanged the term 3D regarding this reference to cyclic-IMS.

We and others have published meanwhile with the use of 4D, frankly without intentional using a trademark, simply to delineate and emphasize the parameters used for resolving and analysing the ions.

Line 119 Normal phase LC should precede MS. **Done**

Line 199 Change text to de novo in italics. **Done**

Figure 1A: Improve labels on the axes. **Done**

Figure 1D: Add a Key for the symbols. Are fragment ions observed in these spectra?

Done. We annotated the fragment ions in the spectra in Figure 1 and all other figures.

Figure 2: Add a key for the symbols. **Done**

Line 270 and throughout the manuscript: Change text to nanoflow replacing nanoLflow. **Done**

Line 295 The reference to Figure 2C should be 3C. **Done**

Lines 323-4 Why does microflow perform better? Is it using higher loading?

Answer: It is not the flow-rate per se or exclusively the flow-rate. We think it is the combination of improved extraction efficiency, resolving power of RP-LC with IMS, instrument and method sensitivity that collectively render a higher coverage of the glycosphingolipidome, even than with nanoflow-LC platform which are expected to render higher sensitivity than microflow LC-MS.

Figure 4A: Add more description in the caption for the MS data panels. **Done**

Lines 380-387 The text describing the quantification strategies is confusing; addition of a table or workflow diagram to Figure 5 would be helpful to the reader.

Answer: we included a diagram and improved the description of the strategies for quantification.

Figure 5: Change the y axis from % to Measured GSLs (%) for clarity. Why was 35% CV chosen? 20% CV is a more typical cutoff. A plot of the CV values for all of the GSLs would also be useful.

Answer:

We added the plot of CV values for all GSLs. We used 35% to compare strategy performance given the limited ISTD available for GSL classes and subclasses and also the fractionation needed, so a bit more permissive cut-off felt fair in this context. However, reviewer can appreciate that many subclasses are less than 30% too (which is still a cut-off widely used in unbiased lipidomics). We are hoping that with the advent of ISTD procurement these cut-offs can be decreased to 20% or less.

Figure 6b: The heat map does not show that some GSLs are systematically different between controls and Parkinson's disease patients; it shows that a couple of individual patients' samples have high expression compared to everyone else.

Answer: It is true that on the original scaling of the heatmap the differences were not visible. We now provided the heatmap on a different scale that shows differences between female and male with PD and in controls. Moreover, we added the statistical evaluation of neutral glycolipids and sulfatides and GSL class-based changes with PD (Figure 8b, c, and d).

Figure 7: The GSL data show overlapping distributions in most cases. Despite strong p values, it does not seem like there is much separation between groups. Addition of the mean or median and standard deviation in each group will help clarify the levels of difference for the reader. Addition of Hellinger or Bhattacharya distance would also improve the ability to show whether the two groups separate from each other.

Answer: We added all the p-Values in the Figure 8, and all the means and SD in Supplementary Data 11. We performed Mann-Whitney U-test which is suitable to determine if there is a significant difference between the distributions of two independent samples. It is particularly appropriate when dealing with ordinal data or when the data may have outliers or non-normal distributions and hence, the Hellinger distance is not inherently inferential as it does not provide a statistical test of significance.

It is evident in Figure 8 that females have a higher group separation between PD and Control, ex GT1 b, GM2 class, GM3 18:1;O2/22:2. There is indeed still some group overlap between PD and controls and that is a feature often observable in lipidomic studies of diseases^{14,15}. This is likely due to the bi- or multi-directionality of synthesis/degradation and subclass lipid conversion as part of metabolic remodeling. Hence, a panel of GSLs with discriminative power between groups and phenotype, as discussed in context of male and female differences is envisaged for further application.

Lines 676 and 686: Do the authors mean SPE cartridges? Please provide the part numbers to support replication of results in other labs. **Done**

Line 703: Define the column dimensions (length x ID) and check particle size microns not mm. **Done**

Lines 764-5 and 780: Are the authors making the libraries of GSLs available to the public? This resource could have very strong value for the community.

Answer: We provide a link in the manuscript for downloading the basic library we use for GSL annotation in biological samples. This library will be available if and upon manuscript acceptance. Spectral library is available at <https://www.unimedizin-mainz.de/lipidomics-unit/lipid-research/research-tools.html> and <https://github.com/dtots17/Lipidomics-Spectral-Database.git> .

The reviewer(s) and readers can however, also appreciate and use the CCS values and RT we provided for all the sialylated GSLs, neutrals and sulfatides in Suppl. Material.

Table 1 is quite long and may need to be summarized with the full version included in the supplement.

Answer: We restructured the Table 1 to encompass the species we detected in serum; where the 4D parameters were moved to Supplementary Table 4, 5, and 6. Given that we included all the 3D-cubic plots of all three classes sialylated, neutral and sulfatides in Figure 5 in the main manuscript we hope this is an appropriate solution for the reader.

Supplemental tables should have the caption and headings on each page in the document, or just submitted as spreadsheets. **Done**

References

1. Merrill, A. H. Sphingolipid and Glycosphingolipid Metabolic Pathways in the Era of Sphingolipidomics. *Chem Rev* **111**, 6387–6422 (2011).
2. Kirsch, S., Müthing, J., Peter-Katalinić, J. & Bindila, L. On-line nano-HPLC/ESI QTOF MS monitoring of α 2-3 and α 2-6 sialylation in granulocyte glycosphingolipidome. *Biol Chem* **390**, 657–672 (2009).
3. Kirsch, S. Ceramide Profiles of Human Serum Gangliosides GM2 and GD1a exhibit Cancer-associated Alterations. *J Glycomics Lipidomics* **01**, (2012).

4. Hořejší, K. *et al.* Comprehensive Identification of Glycosphingolipids in Human Plasma Using Hydrophilic Interaction Liquid Chromatography—Electrospray Ionization Mass Spectrometry. *Metabolites* **11**, 140 (2021).
5. Wang, Z. & Xia, Y. Selective Enrichment via TiO₂ Magnetic Nanoparticles Enables Deep Profiling of Circulating Neutral Glycosphingolipids. *Anal Chem* **96**, 16955–16963 (2024).
6. Lerner, R. *et al.* Four-dimensional trapped ion mobility spectrometry lipidomics for high throughput clinical profiling of human blood samples. *Nat Commun* **14**, 937 (2023).
7. Hořejší, K. *et al.* Comprehensive characterization of complex glycosphingolipids in human pancreatic cancer tissues. *Journal of Biological Chemistry* **299**, 102923 (2023).
8. MacGibbon, A. K. H. Composition and Structure of Bovine Milk Lipids. in *Advanced Dairy Chemistry, Volume 2* 1–32 (Springer International Publishing, Cham, 2020). doi:10.1007/978-3-030-48686-0_1.
9. Salcedo, J. *et al.* Application of industrial treatments to donor human milk: influence of pasteurization treatments, storage temperature, and time on human milk gangliosides. *NPJ Sci Food* **2**, 5 (2018).
10. Hewelt-Belka, W., Młynarczyk, M., Garwolińska, D. & Kot-Wasik, A. Characterization of GM3 Gangliosides in Human Milk throughout Lactation: Insights from the Analysis with the Use of Reversed-Phase Liquid Chromatography Coupled to Quadrupole Time-Of-Flight Mass Spectrometry. *J Agric Food Chem* **71**, 17899–17908 (2023).
11. Hakomori, S. The glycosynapse. *Proceedings of the National Academy of Sciences* **99**, 225–232 (2002).
12. Samraj, A. N. *et al.* A red meat-derived glycan promotes inflammation and cancer progression. *Proceedings of the National Academy of Sciences* **112**, 542–547 (2015).
13. Yin, J. *et al.* Hypoxic Culture Induces Expression of Sialin, a Sialic Acid Transporter, and Cancer-Associated Gangliosides Containing Non-Human Sialic Acid on Human Cancer Cells. *Cancer Res* **66**, 2937–2945 (2006).
14. Huynh, K. *et al.* High-Throughput Plasma Lipidomics: Detailed Mapping of the Associations with Cardiometabolic Risk Factors. *Cell Chem Biol* **26**, 71-84.e4 (2019).

15. Cífková, E. *et al.* Lipidomic differentiation between human kidney tumors and surrounding normal tissues using HILIC-HPLC/ESI-MS and multivariate data analysis. *Journal of Chromatography B* **1000**, 14–21 (2015).

Answer to REVIEWER COMMENTS

Reviewer #1 (Remarks to the Author):

We have thoroughly reviewed the authors' responses and appreciate their excellent work during the revision process. Their efforts have not only addressed the feedback effectively but also further enhanced the quality of manuscript. We have no additional comments and firmly believe that this work represents a valuable contribution to the field of glycosphingolipid analysis.

Answer: We highly appreciate your feedback, the constructive and detailed review of our work, which contributed to significant improvement of the manuscript quality.

Reviewer #2 (Remarks to the Author):

Answer: We highly appreciate your constructive feedback throughout the revision process.

Reviewer #3 (Remarks to the Author):

Answer: We highly appreciate your constructive feedback throughout the revision process.

Reviewer #4 (Remarks to the Author):

The revisited paper is carefully addressed to each inquiry and question. I think this paper will be very useful in the GSL research community.

Answer: We highly appreciate your feedback, the constructive review of our work, which significantly improved the quality of the manuscript.

Reviewer #5 (Remarks to the Author):

The authors have addressed the questions raised in the first review and significantly improved their manuscript. The text and methods are clear now and should aid in other investigators applying the same method. The different mechanisms of data sharing are also strong. All concerns have been appropriately addressed with thorough and well thought out answers, and the corresponding changes have been made in the manuscript.

Only one minor recommendation, add a key for the sugar residues in the figure or caption for Supplemental Figures 2, 5, and 6.

Answer: We thank reviewer for the constructive feedback and useful suggestion. Following this suggestion of the reviewer and in agreement with the editor's suggestion we included now Figure 1a containing the symbols and their structural assignment. We also provided symbol description in each figure caption.